# FACTORIZED CONTRASTIVE LEARNING: Going Beyond Multi-view Redundancy

**Paul Pu Liang**[1*], **Zihao Deng**[2*], **Martin Q. Ma**[1*]
**James Zou**[3], **Louis-Philippe Morency**[1], **Ruslan Salakhutdinov**[1]
[1]Carnegie Mellon University, [2]University of Pennsylvania, [3]Stanford University
`pliang@cs.cmu.edu,zihaoden@cs.cmu.edu,qianlim@cs.cmu.edu`

## Abstract

In a wide range of multimodal tasks, contrastive learning has become a particularly appealing approach since it can successfully learn representations from abundant unlabeled data with only pairing information (e.g., image-caption or video-audio pairs). Underpinning these approaches is the assumption of *multi-view redundancy* - that shared information between modalities is necessary and sufficient for downstream tasks. However, in many real-world settings, task-relevant information is also contained in modality-unique regions: information that is only present in one modality but still relevant to the task. How can we learn self-supervised multimodal representations to capture both shared and unique information relevant to downstream tasks? This paper proposes FACTORCL, a new multimodal representation learning method to go beyond multi-view redundancy. FACTORCL is built from three new contributions: (1) factorizing task-relevant information into shared and unique representations, (2) capturing task-relevant information via maximizing MI lower bounds and removing task-irrelevant information via minimizing MI upper bounds, and (3) multimodal data augmentations to approximate task relevance without labels. On large-scale real-world datasets, FACTORCL captures both shared and unique information and achieves state-of-the-art results on six benchmarks.

## 1 Introduction

Learning representations from different modalities is a central paradigm in machine learning [48]. Today, a popular learning method is to first pre-train general representations on unlabeled multimodal data before fine-tuning on task-specific labels [10, 40, 47, 48, 50]. These current multimodal pre-training approaches have largely been inherited from prior work in multi-view learning [13, 58] that exploit a critical assumption of *multi-view redundancy*: the property that shared information between modalities is almost exactly what is relevant for downstream tasks [70, 73, 76]. When this assumption holds, approaches based on contrastive pre-training to capture shared information [13, 41, 61, 72], followed by fine-tuning to keep task-relevant shared information [76], have seen successful applications in learning from images and captions [61], video and audio [3], speech and transcribed text [58], and instructions and actions [21]. However, our paper studies two fundamental limitations in the application of contrastive learning (CL) to broader real-world multimodal settings (see Figure 1 for a visual depiction and experimental results showing the performance drop of CL):

1. **Low *shared* information** relevant to tasks: There exists a wide range of multimodal tasks involving small amounts of shared information, such as between cartoon images and figurative captions (i.e., not literal but metaphoric or idiomatic descriptions of the images [52, 88]). In these situations, standard multimodal CL will only receive a small percentage of information from the learned representations and struggle to learn the desired task-relevant information.
2. **High *unique* information** relevant to tasks: Many real-world modalities can provide unique information not present in other modalities. Examples include healthcare with medical sensors or robotics with force sensors [45, 49]. Standard CL will discard task-relevant unique information, leading to poor downstream performance.

---

[*]First three authors contributed equally.

37th Conference on Neural Information Processing Systems (NeurIPS 2023).

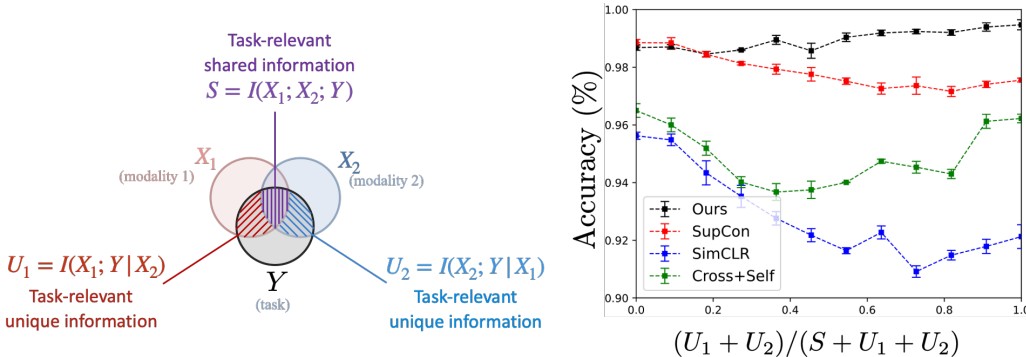

Figure 1: **Left**: We define $S = I(X_1; X_2; Y)$ as task-relevant shared information and $U_1 = I(X_1; Y|X_2)$, $U_2 = I(X_2; Y|X_1)$ as task-relevant unique information. **Right**: On controllable datasets with varying ratios of $S$, $U_1$, and $U_2$, standard CL captures $S$ but struggles when there is more $U_1$ and $U_2$. Our FACTORCL approach maintains best performance, whereas SimCLR [13] and SupCon [41] see performance drops as unique information increases, and Cross+Self [33, 36, 44, 89] recovers in fully unique settings but suffers at other ratios.

In light of these limitations, how can we design suitable multimodal learning objectives that work beyond multi-view redundancy? In this paper, starting from the first principles in information theory, we provide formal definitions of shared and unique information via conditional mutual information and propose an approach, FACTORIZED CONTRASTIVE LEARNING (FACTORCL for short), to learn these multimodal representations beyond multi-view redundancy using three key ideas. The first idea is to explicitly *factorize* shared and unique representations. The second idea is to *capture task-relevant* information via maximizing lower bounds on MI and *remove task-irrelevant* information via minimizing upper bounds on MI, resulting in representations with sufficient and necessary information content. Finally, a notion of task relevance without explicit labels in the self-supervised setting is achieved by leveraging *multimodal augmentations*. Experimentally, we evaluate the effectiveness of FACTORCL on a suite of synthetic datasets and large-scale real-world multimodal benchmarks involving images and figurative language [88], prediction of human sentiment [91], emotions [93], humor [27], and sarcasm [12], as well as patient disease and mortality prediction from health indicators and sensor readings [38], achieving new state-of-the-art performance on six datasets. Overall, we summarize our key technical contributions here:

1. A new analysis of contrastive learning performance showing that standard multimodal CL fails to capture task-relevant unique information under low shared or high unique information cases.
2. A new contrastive learning algorithm called FACTORCL:
   (a) FACTORCL factorizes task-relevant information into shared and unique information, expanding contrastive learning to better handle low shared or high unique information.
   (b) FACTORCL optimizes shared and unique information separately, by removing task-irrelevant information via MI upper bounds and capturing task-relevant information via lower bounds, yielding optimal task-relevant representations.
   (c) FACTORCL leverages multimodal augmentations to approximate task-relevant information, enabling self-supervised learning from our proposed FACTORCL.

## 2 Analysis of Multi-view Contrastive Learning

We begin by formalizing definitions of four types of information: shared, unique, task-relevant, and task-irrelevant information in multimodal data. To formalize the learning setting, we assume there exist two modalities expressed as random variables $X_1$ and $X_2$ with outcomes $x_1$ and $x_2$, and a task with the random variable $Y$ and outcome $y$. We denote $X_{-i}$ as the other modality where appropriate.

**Shared and unique information**: We formalize shared and unique information by decomposing the total multimodal information $I(X_1, X_2; Y)$ into three conditional mutual information (MI) terms:

$$I(X_1, X_2; Y) = \underbrace{I(X_1; X_2; Y)}_{S \text{ = shared}} + \underbrace{I(X_1; Y|X_2)}_{U_1 \text{ = uniqueness in } X_1} + \underbrace{I(X_2; Y|X_1)}_{U_2 \text{ = uniqueness in } X_2} , \qquad (1)$$

where $I(X_1, X_2; Y) = \int p(x_1, x_2, y) \log \frac{p(x_1, x_2, y)}{p(x_1, x_2)p(y)} dx_1 dx_2 dy$ is the total MI between the joint random variable $X_1, X_2$ and the task $Y$, $S = I(X_1; X_2; Y) = I(X_1; X_2) - I(X_1; X_2|Y) = \int p(x_1, x_2) \log \frac{p(x_1, x_2)}{p(x_1)p(x_2)} dx_1 dx_2 - I(X_1; X_2|Y)$ is the task-relevant shared in-

formation, $I(X_1; X_2|Y) = \int p(x_1, x_2, y) \log \frac{p(x_1, x_2|y)}{p(x_1|y)p(x_2|y)} dx_1 dx_2 dy$ is the task-irrelevant shared information, and $U_1 = I(X_1; Y|X_2)$, $U_2 = I(X_2; Y|X_1)$ denote unique task-relevant information.

**Limitations of CL**: Current approaches for CL maximize mutual information $I(X_1; X_2)$ (and subsequently task-relevant shared information $I(X_1; X_2; Y)$ during supervised fine-tuning), without modeling unique information. These methods generally learn a pair of representations [73, 76],

$$Z_1 = \underset{Z_1 := f_\theta(X_1)}{\arg\max} \, I(Z_1; X_2), \quad Z_2 = \underset{Z_2 := f_\theta(X_2)}{\arg\max} \, I(X_1; Z_2). \tag{2}$$

For example, $Z_1$ could encode images $X_1$ and $Z_2$ encodes text $X_2$ via maximizing a lower bound on $I(X_1; X_2)$ using the NCE objective [58]. The NCE objective falls into a broader class of contrastive learning methods [13, 15, 28, 41, 61] that model the ratio between joint densities $p(x_1, x_2)$ and product of marginal densities $p(x_1)p(x_2)$ using positive and negative samples [57, 59, 60, 79, 84] or probabilistic classifiers [55, 77]. It has been shown that contrastive learning works well under the assumption of multi-view redundancy [4, 31, 70, 71, 76]:

**Definition 1.** *(Multi-view redundancy)* $\exists \epsilon > 0$ *such that* $I(X_1; Y|X_2) \le \epsilon$ *and* $I(X_2; Y|X_1) \le \epsilon$.

In other words, the task-relevant information in data is mostly shared across both views and the unique information is at most a small $\epsilon$. From a representation perspective, Tian et al. [72] further introduces the assumption that the optimal representation is minimal and sufficient, where all learned task-relevant information is shared information: $I(Z_1; Y|X_2) = I(Z_2; Y|X_1) = 0$. While the multi-view redundancy is certainly true for particular types of multimodal distributions, it crucially ignores settings that display *multi-view non-redundancy* and unique information can be important, such as when health indicators, medical sensors, and robotic visual or force sensors each provide unique information not present in other modalities [45, 49].

**Definition 2.** *(Multi-view non-redundancy)* $\exists \epsilon > 0$ *such that* $I(X_1; Y|X_2) > \epsilon$ *or* $I(X_2; Y|X_1) > \epsilon$.

Under multi-view non-redundancy, we show that standard CL only receives a weak training signal since it can only maximize a lower bound on shared information $I(X_1; X_2)$, and struggles to learn task-relevant unique information. We formalize this intuition with the following statement:

**Theorem 1.** *(Suboptimality of standard CL) When there is multi-view non-redundancy as in Definition 2, given optimal representations $\{Z_1, Z_2\}$ that satisfy Eq.(2 and $I(Z_1; Y|X_2) = I(Z_2; Y|X_1) = 0$ [72], we have that*

$$I(Z_1, Z_2; Y) = I(X_1, X_2; Y) - I(X_1; Y|X_2) - I(X_2; Y|X_1) = I(X_1; X_2) - I(X_1; X_2|Y) < I(X_1, X_2; Y). \tag{3}$$

*Correspondingly, the Bayes error rate $P_e(Z_1, Z_2) := 1 - \mathbb{E}_{p(z_1, z_2)} \left[ \max_{y \in Y} P\left(\hat{Y} = y \mid z_1, z_2\right)\right]$ of contrastive representations $\{Z_1, Z_2\}$ for a downstream task $Y$ is given by:*

$$P_e \le 1 - \exp\left[I(X_1, X_2; Y) - I(X_1; Y|X_2) - I(X_2; Y|X_1) - H(Y)\right] \tag{4}$$
$$= 1 - \exp\left[I(X_1; X_2; Y) - H(Y)\right] \tag{5}$$

We include proofs and a detailed discussion of the assumptions in Appendix B. Based on Eq.(3), $I(Z_1, Z_2; Y)$ decreases with higher task-relevant unique information $I(X_1; Y|X_2)$ and $I(X_2; Y|X_1)$; we call this the difference $I(X_1, X_2; Y) - I(Z_1, Z_2; Y)$ the *uniqueness gap*. The uniqueness gap measures the loss in task-relevant information between the input and encoded representation: as task-relevant unique information grows, the uniqueness gap increases. In addition, $I(Z_1, Z_2; Y)$ also drops with lower $I(X_1; X_2)$ (i.e., two modalities sharing little information to begin with), or with higher $I(X_1; X_2|Y)$ (i.e., when the shared information is mostly task-irrelevant). Similarly, in Eq.(5), the Bayes error rate of using $\{Z_1, Z_2\}$ for prediction is directly related to the task-relevant information in $\{Z_1, Z_2\}$: error on the downstream task increases with higher unique information and lower shared information.

## 3 FACTORIZED CONTRASTIVE LEARNING

We now present a suite of new CL objectives that alleviate the challenges above and work at all ranges of shared and unique information. At a high level, we aim to learn a set of factorized representations $Z_{S_1}, Z_{S_2}, Z_{U_1}, Z_{U_2}$ representing task-relevant information in $X_1$ shared with $X_2$, in $X_2$ shared with $X_1$, unique to $X_1$, and unique to $X_2$ respectively. As common in practice [61, 72], we define

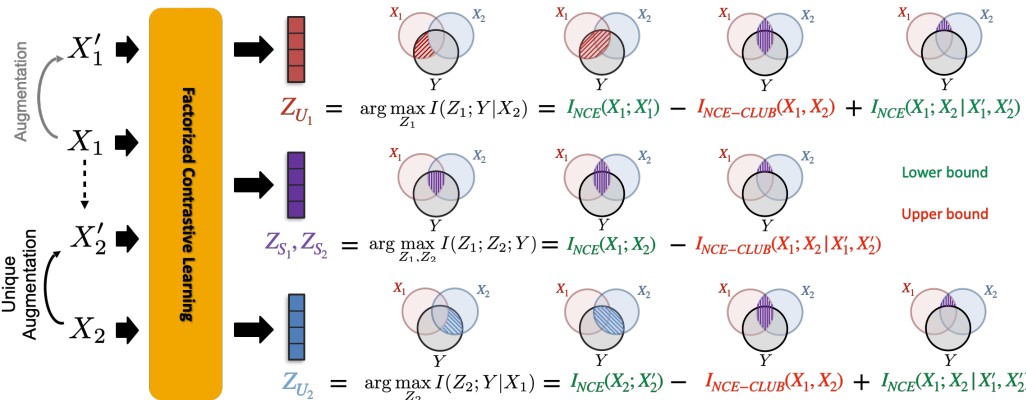

Figure 2: FACTORCL: We propose a self-supervised CL method to learn *factorized* representations $Z_{S_1}$, $Z_{S_2}$, $Z_{U_1}$, and $Z_{U_2}$ to capture task-relevant information shared in both $X_1$ and $X_2$, unique to $X_1$, and unique to $X_2$. By starting with information-theoretic first principles of shared and unique information, we design contrastive estimators to both *capture task-relevant* and *remove task-irrelevant* information, where a notion of task-relevance without explicit labels is afforded by a new definition of *multimodal augmentations* $X_1'$, $X_2'$. Lower bounds are in green and upper bounds are in red.

neural networks $f_\theta$ with trainable parameters $\theta$ to extract representations from inputs $X_1$ and $X_2$. Learning these parameters requires optimizing differentiable and scalable training objectives to capture task-relevant shared and unique information (see overview in Figure 2):

$$Z_{S_1} = \operatorname*{arg\,max}_{Z_1=f_\theta(X_1)} I(Z_1; X_2; Y), \qquad Z_{S_2} = \operatorname*{arg\,max}_{Z_2=f_\theta(X_2)} I(Z_2; X_1; Y), \qquad (6)$$

$$Z_{U_1} = \operatorname*{arg\,max}_{Z_1=f_\theta(X_1)} I(Z_1; Y|X_2), \qquad Z_{U_2} = \operatorname*{arg\,max}_{Z_2=f_\theta(X_2)} I(Z_2; Y|X_1). \qquad (7)$$

where $I(Z_1; X_2; Y) = I(Z_1; X_2) - I(Z_1; X_2|Y)$ is the shared information and $I(Z_2; X_1; Y) = I(Z_2; X_2) - I(Z_2; X_1|Y)$ is the unique information. One important characteristic of our framework is that when unique information is zero: $I(X_1; Y|X_2) = 0$ and $I(X_2; Y|X_1) = 0$, or all shared information is task-relevant: $I(X_1; X_2; Y) = I(X_1; X_2)$, our framework recovers standard CL as in Eq.(2). However, as we have previously indicated and will show empirically, these assumptions can easily be violated, and our framework enlarges Eq.(2) to cases where unique information is present.

The learned $Z$s can then be used as input to a linear classifier and fine-tuned to predict the label for multimodal classification or retrieval tasks. However, the shared and unique MI terms above are often intractable in practice. In the next section, we will build up our method step by step, eventually showing that each term in Eqs.(6- 7) can be approximated as follows:

$$S = I(X_1; X_2; Y) \geq I_{\text{NCE}}(X_1; X_2) - I_{\text{NCE-CLUB}}(X_1; X_2|X_1', X_2') \qquad (8)$$

$$U_i = I(X_i; Y|X_{-i}) \geq I_{\text{NCE}}(X_i; X_i') - I_{\text{NCE-CLUB}}(X_1; X_2) + I_{\text{NCE}}(X_1; X_2|X_1', X_2') \qquad (9)$$

where $I_{\text{NCE}}$ and $I_{\text{NCE-CLUB}}$ are scalable contrastive estimators (Section 3.1) and $X_1'$, $X_2'$ are suitable data augmentations (Section 3.2) on each modality. Overall, these equations can be interpreted as both positive and negative signals to learn representations for $S$ and $U$. For shared information $S$, the estimator maximizes task-relevant shared information via $I_{\text{NCE}}(X_1; X_2)$ while removing task-irrelevant shared information via a novel upper bound $-I_{\text{NCE-CLUB}}(X_1; X_2|X_1', X_2')$. For unique information $U_i$, we capture task-relevant uniqueness via $+I_{\text{NCE}}(X_i; X_i')$ while non-unique information is removed via $-(I_{\text{NCE-CLUB}}(X_1; X_2) - I_{\text{NCE}}(X_1; X_2|X_1', X_2'))$. In the following sections, we derive this final objective step-by-step: (1) approximating the MI objectives in $S$ and $U$ with CL estimators, (2) relaxing the dependence on labels $Y$ with self-supervised data augmentations, finally (3) discussing overall training and implementation details of end-to-end self-supervised learning.

### 3.1 Supervised FACTORCL with shared and unique information

To capture shared and unique information via an objective function, we will need to maximize lower bounds for all terms with a positive sign in Eq.(8) and (9) ($I(X_1; X_2), I(X_i; Y), I(X_1; X_2|Y)$) and minimize upper bounds for all terms with a negative sign ($I(X_1; X_2), I(X_1; X_2|Y)$). Our first theorem derives general lower and upper bounds for MI terms as variants of contrastive estimation:

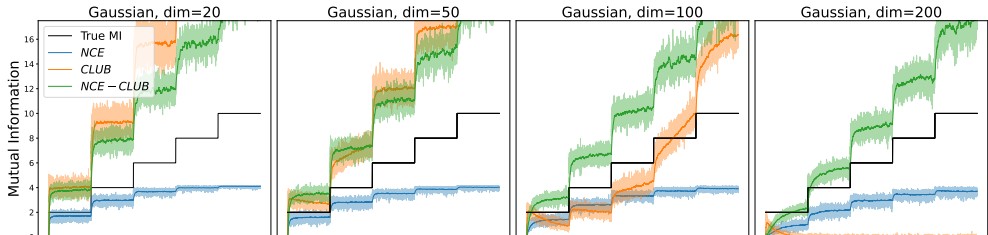

Figure 3: Estimated $I_{\text{NCE}}$ lower bound [58] and our proposed upper bound $I_{\text{NCE-CLUB}}$ on sample distributions with changing mutual information: our upper bound is tighter, more accurate, and more stable than $I_{\text{CLUB}}$ upper bound [16], and also comes for 'free' via jointly estimating both lower and upper bounds simultaneously. We find that as dimension increases, the $I_{\text{CLUB}}$ estimator collapses to zero and no longer tracks true MI.

**Theorem 2.** *(Contrastive estimators for $I(X_1; X_2)$) Defining the NCE and NCE-CLUB estimators,*

$$I_{\text{NCE}}(X_1; X_2) = \mathbb{E}_{\substack{x_1, x_2^+ \sim p(x_1, x_2) \\ x_2^- \sim p(x_2)}} \left[ \log \frac{\exp f(x_1, x_2^+)}{\sum_k \exp f(x_1, x_2^-)} \right] \tag{10}$$

$$I_{\text{NCE-CLUB}}(X_1; X_2) = \mathbb{E}_{x_1, x_2^+ \sim p(x_1, x_2)} \left[ f^*(x_1, x_2^+) \right] - \mathbb{E}_{\substack{x_1 \sim p(x_1) \\ x_2^- \sim p(x_2)}} \left[ f^*(x_1, x_2^-) \right] \tag{11}$$

*where $f^*(x_1, x_2)$ is the optimal critic from $I_{\text{NCE}}$ plugged into the $I_{\text{CLUB}}$ objective [16]. We call the proposed plug-in objective Eq.(11) $I_{\text{NCE-CLUB}}$, and obtain lower and upper bounds on $I(X_1; X_2)$:*

$$I_{\text{NCE}}(X_1; X_2) \leq I(X_1; X_2) \leq I_{\text{NCE-CLUB}}(X_1; X_2). \tag{12}$$

*Proof.* The lower bound $I_{\text{NCE}}(X_1; X_2) \leq I(X_1; X_2)$ follows from Oord et al. [58]: optimizing the objective leads to an optimal critic [60] $f^* = \log p(x_1|x_2) + c(x_1)$, with a deterministic function $c(\cdot)$. Plugging optimal critic $f^*$ into $I_{\text{NCE-CLUB}}(X_1; X_2)$ cancels out the $c(x_1)$ term and yields $I_{\text{NCE-CLUB}}(X_1; X_2)$ and $I(X_1; X_2) \leq I_{\text{NCE-CLUB}}$. We include a detailed proof in Appendix C.1. $\quad\square$

$I_{\text{NCE-CLUB}}(X_1; X_2)$ gives a desired upper bound of $I(X_1; X_2)$ "for free" while avoiding separately optimizing lower bound and upper bounds. In Figure 3, we show these two bounds in practice across two Gaussian distributions $X_1$ and $X_2$ with varying amounts of MI $I(X_1; X_2)$. We use the second formulation of $I_{\text{CLUB}}$ [16], which assumes $p(x_1|x_2)$ to be unknown. Our upper bound is empirically tighter (see Figure 3) and comes for "free" via jointly maximizing the lower bound $I_{\text{NCE}}$. These lower and upper bounds can be seen as new contrastive objectives over positive and negative $(x_1, x_2)$ pairs, enabling a close integration with existing pre-training paradigms. Finally, we can similarly obtain bounds for the conditional MI $I_{\text{NCE}}(X_1; X_2|Y) \leq I(X_1; X_2|Y) \leq I_{\text{NCE-CLUB}}(X_1; X_2|Y)$:

$$I_{\text{NCE}}(X_1; X_2|Y) = \mathbb{E}_{p(y)} \left[ \mathbb{E}_{\substack{x_1, x_2^+ \sim p(x_1, x_2|y) \\ x_2^- \sim p(x_2|y)}} \left[ \log \frac{\exp f(x_1, x_2^+, y)}{\sum_k \exp f(x_1, x_2^-, y)} \right] \right] \tag{13}$$

$$I_{\text{NCE-CLUB}}(X_1; X_2|Y) = \mathbb{E}_{p(y)} \left[ \mathbb{E}_{x_1, x_2^+ \sim p(x_1, x_2|y)} \left[ f^*(x_1, x_2^+, y) \right] - \mathbb{E}_{\substack{x_1 \sim p(x_1|y) \\ x_2^- \sim p(x_2|y)}} \left[ f^*(x_1, x_2^-, y) \right] \right] \tag{14}$$

These two bounds result in *conditional CL* objectives [51, 74, 78] - they differ critically from standard CL methods since they capture task-irrelevant shared information that remains between $X_1$ and $X_2$ after observing $Y$. This task-irrelevant shared information is removed by minimizing its upper bound. Note that $f(x_1, x_2, y)$ here denotes a different function from $f(x_1, x_2)$ in Eq.(10), as the general forms are different (taking in $x_1, x_2$ versus $x_1, x_2, y$). $f(x_1, x_2, y)$ can be implemented in different ways, e.g., $g([x_1, y])^T h(x_2)$ where $g(), h()$ are trainable encoders and $[x_1, y]$ denotes concatenation [69].

### 3.2 Self-supervised FACTORCL via multimodal augmentations

The derivations above bring about supervised CL objectives with access to $Y$ [41]. For unsupervised CL [58, 72], we derive similar objectives without access to $Y$ by leveraging semantic augmentations on each modality. Denote $X'$ as some augmentation of $X$ (e.g., rotating, shifting, or cropping). Under

the *optimal augmentation* assumption from Tian et al. [72] (restated below), replacing $Y$ with $X'$ in our formulations enables learning of task-relevant information without access to labels:

**Definition 3.** *(Optimal unimodal augmentation) [72]* $X_1'$ *is an optimal unimodal augmentation for* $X_1$ *when* $I(X; X') = I(X; Y)$*, which implies that the only information shared between* $X$ *and* $X'$ *is task-relevant with no irrelevant noise.*

This assumption is satisfied when all information shared between $X$ and $X'$ is task-relevant, which implies that the augmentation keeps task-relevant information constant while changing task-irrelevant information. In the case of image classification, task-relevant information is the object in the picture, while task-irrelevant information is the background. By performing two separate unimodal augmentations giving $X_1'$ and $X_2'$, we can substitute contrastive estimators in Eqs.(13) and (14), by replacing $I(X_i; Y)$ terms with $I(X_i; X_i')$ and replacing $I(X_1; X_2|Y)$ terms with $I(X_1; X_2|X_1', X_2')$:

$$I_{\text{NCE}}(X_1; X_2|X_1', X_2') = \mathbb{E}_{p(x_1', x_2')}\left[\mathbb{E}_{\substack{x_1, x_2^+ \sim p(x_1, x_2|x_1', x_2') \\ x_2^- \sim p(x_2|x_1', x_2')}}\left[\log \frac{\exp f(x_1, x_2^+, x_1', x_2')}{\sum_k \exp f(x_1, x_2^-, x_1', x_2')}\right]\right] \quad (15)$$

$$I_{\text{NCE-CLUB}}(X_1; X_2|X_1', X_2') = \mathbb{E}_{p(x_1', x_2')}\Big[\mathbb{E}_{x_1, x_2^+ \sim p(x_1, x_2|x_1', x_2')}[f^*(x_1, x_2^+, x_1', x_2')]$$
$$- \mathbb{E}_{\substack{x_1 \sim p(x_1|x_1', x_2') \\ x_2^- \sim p(x_2|x_1', x_2')}}[f^*(x_1, x_2^-, x_1', x_2')]\Big] \quad (16)$$

The objectives can be seen as conditional contrastive learning on augmentations $(X_1', X_2')$. Here again $f(x_1, x_2, x_1', x_2')$ is different from the critics in Eqs.(13 because of the different general forms. We implement $f()$ here as $g([x_1, x_1'])^T h([x_2, x_2'])$ where $g(), h()$ are trainable encoders specific for each modality and $[x_1, x_1']$ denotes concatenation. This concatenation is justified by the CMI estimators in Sordoni et al. [69], who show that concatenating the conditioning variable with the input in the critic $f(x_1, x_2, x_1', x_2')$ yields a Conditional InfoNCE estimator (Eq.(15)) that is a lower bound for CMI. However, the exact Conditional InfoNCE estimator learns a different conditional distribution $p(x_1, x_2|x_1', x_2')$ for each augmented pair $x_1', x_2'$, which can be prohibitively expensive. We could approximate this by creating multiple augmentations of a single paired $x_1, x_2$. Our code uses one augmented pair $x_1', x_2'$ for each $x_1, x_2$ but could be extended to multiple pairs, and we find this simple approach yields consistent CMI lower and upper bounds that are empirically comparable to existing CMI estimators [55, 69]. We include full comparisons and implementation details in Appendix D.1, and in Appendix C.2 we discuss an alternative interpretation based on viewing CL as kernel learning which permits using conditional kernel estimation for our objectives.

Although we find this method to work well in practice, a more careful analysis reveals that 2 separate unimodal augmentations $X_1'$ and $X_2'$ each satisfying $I(X_i; X_i') = I(X_i; Y)$ do not together satisfy $I(X_1; X_2|Y) = I(X_1; X_2|X_1', X_2')$ needed for the substitution in Eqs.(15) and (16) to hold with equality. To satisfy this property exactly, we define optimal multimodal augmentations:

**Definition 4.** *(Optimal multimodal augmentation)* $X_1'$ *and* $X_2'$ *are optimal multimodal augmentation for* $X_1$ *and* $X_2$ *when* $I(X_1, X_2; X_1', X_2') = I(X_1, X_2; Y)$*, which implies that the only information shared between* $X_1, X_2$ *and* $X_1', X_2'$ *is task-relevant with no irrelevant noise.*

We satisfy $I(X_1, X_2; X_1', X_2') = I(X_1, X_2; Y)$ using two steps:

$$\text{Unimodal aug: } X_1' \text{ s.t. } I(X_1; X_1') = I(X_1; Y), \quad (17)$$
$$\text{Unique aug: } X_2' \text{ s.t. } I(X_2; X_2'|X_1) = I(X_2; Y|X_1). \quad (18)$$

We call the second step *unique augmentation*: after observing $X_1$, we create augmented $X_2'$ from $X_2$ to keep task-relevant information not already in $X_1$. To empirically satisfy optimal multimodal augmentations, we avoid augmentations in one modality that will remove or strongly destroy information shared with the other modality. For example, in image captioning, we should avoid image augmentations such as cropping that destroy information from the caption (e.g., cropping object parts referred to by the caption), and instead, only augment images via flipping or color jittering which retains all caption information. Figure 4 shows an example of unique augmentation that satisfies these conditions. In our experiments, we will show that our augmentations consistently perform better than standard augmentations (Table 3), suggesting that approximately satisfying Eqs.(17) and (18) can be empirically sufficient, which is simple and straightforward to implement on real-world datasets.

### 3.3 Overall method and implementation

| **Algorithm 1** Standard multimodal CL. | **Algorithm 2** FACTORCL. |
|---|---|
| **Require:** Multimodal dataset $\{\mathbf{X_1}, \mathbf{X_2}\}$. | **Require:** Multimodal dataset $\{\mathbf{X_1}, \mathbf{X_2}\}$. |

<div>

Initialize networks $f(\cdot)$.
**while** not converged **do**
  **for** sampled batch $\{\boldsymbol{x}_1, \boldsymbol{x}_2\}$ **do**
    Estimate $I_{\mathrm{NCE}}(X_1; X_2)$ from Eq. 10
    $\mathcal{L} = -I_{\mathrm{NCE}}(X_1; X_2)$
    Update $f(\cdot)$ to minimize $\mathcal{L}$
  **end for**
**end while**
**return** $f(\cdot)$

</div>

<div>

Initialize networks $f(\cdot)$.
**while** not converged **do**
  **for** sampled batch $\{\boldsymbol{x}_1, \boldsymbol{x}_2\}$ **do**
    $\boldsymbol{x}_1' \leftarrow$ **Augment**$(\boldsymbol{x}_1)$
    $\boldsymbol{x}_2' \leftarrow$ **Unique-Augment**$(\boldsymbol{x}_2|\boldsymbol{x}_1)$
    Plug $\boldsymbol{x}_1'$ and $\boldsymbol{x}_2'$ into Eq. 15 and 16
    Estimate $\mathbf{S}, \mathbf{U_1}, \mathbf{U_2}$ from Eq. 8 and 9
    $\mathcal{L} = -(\mathbf{S} + \mathbf{U_1} + \mathbf{U_2})$
    Update $f(\cdot)$ to minimize $\mathcal{L}$
  **end for**
**end while**
**return** $f(\cdot)$

</div>

Figure 4: Standard vs. unique augmentations for the figurative language [88] dataset. After augmenting text modality $X_1$ independently (same for both augmentation types), we illustrate their differences for image augmentation: unique augmentation on images should avoid removing information referred to by $X_1$ (the text). The text mentions that the car is fast so unique augmentation for images should *not* remove the highway pixels of the image which can suggest the car is fast.

The final algorithm sketch is in Algorithm 2, which we compare against standard CL in Algorithm 1. It can be shown that FACTORCL learns all the task-relevant information from both modalities:

**Theorem 3.** *(Optimality of* FACTORCL*) If* $Z_{S_1}, Z_{S_2}, Z_{U_1}, Z_{U_2}$ *perfectly maximize Eqs.(6-7) and the estimations in Eqs.(8) and (9) are tight, we obtain* $I(X_1, X_2; Y) = I(Z_{S_1}; Z_{S_2}; Y) + I(Z_{U_1}; Y|Z_{S_2}) + I(Z_{U_2}; Y|Z_{S_1})$, *suggesting that* FACTORCL *learns both shared and unique task-relevant information.*

We include the full proof in Appendix C.3. In practice, while we do not expect perfect estimation of MI quantities and maximization with respect to MI objectives, we include implementation details regarding architectures and contrastive objectives that improve empirical performance in Appendix D.1.

**Complexity**: Compared to heuristic combinations of cross-modal and single-modality CL [33, 36, 44, 64, 81, 85, 89], our approach does not significantly increase complexity: (1) upper bounds on MI can be estimated "for free" by directly plugging in the optimal critic from $I_{\mathrm{NCE}}$, (2) removal of task-irrelevant information via $I(X_1; X_2 | X_1', X_2')$ shares encoders with $I_{\mathrm{NCE}}$, and (3) separate unimodal augmentations perform empirically well. We describe some extensions of other self-supervised methods in Appendix C.4.

## 4 Experiments

We run comprehensive experiments on a suite of synthetic and large-scale real-world datasets with varying requirements of shared and unique task-relevant information, comparing our FACTORCL method to key baselines:

1. SimCLR [13]: the straightforward method of cross-modal $(X_1, X_2)$ contrastive learning.
2. Cross+Self [33, 36, 44, 64, 85, 89]: captures a range of methods combining cross-modal $(X_1, X_2)$ CL with additional unimodal $(X_i, X_i')$ CL objectives. This category also includes other ways of preserving unique information, such as through (variational) autoencoder reconstructions [81].
3. Cross+Self+Fact [86, 89]: A factorized extension of Cross+Self, which is approximately done in prior work that adds separate (typically pre-trained) unimodal encoders for each modality.
4. SupCon [41], which learns $I(X_1; X_2 | Y)$ using CL conditioned on $Y$ from labeled data.

We also carefully ablate each component of our method and investigate factors, including training data size and choice of augmentations. The intermediate ablations that emerge include:

1. FACTORCL-SUP: The supervised CL version which uses labels $Y$ in Eqs.(13) and (14).

Table 1: We probe whether contrastive representations learned by classic CL methods and FACTORCL contain shared $w_s$ or unique $w_1, w_2$ information. FACTORCL captures the most unique information.

| Model | SimCLR | | Cross+self | | SupCon | | FACTORCL | | | |
|---|---|---|---|---|---|---|---|---|---|---|
| Representations | $Z_1$ | $Z_2$ | $Z_1$ | $Z_2$ | $Z_1$ | $Z_2$ | $Z_{U_1}$ | $Z_{U_2}$ | $Z_{S_1}$ | $Z_{S_2}$ |
| $I(Z; w_1)$ | 4.45 | 0.16 | 4.39 | 0.14 | 5.17 | 0.19 | **7.83** | 0.03 | 6.25 | 0.04 |
| $I(Z; w_2)$ | 0.17 | 3.92 | 0.13 | 4.26 | 0.23 | 5.17 | 0.06 | **7.17** | 0.05 | 5.79 |
| $I(Z; w_s)$ | 12.61 | 12.06 | 11.30 | 11.47 | 7.48 | 7.17 | 9.47 | 9.89 | 10.13 | 9.40 |

2. FACTORCL-SSL: The fully self-supervised version of our approach replacing $Y$ with multimodal augmentations $X_1'$ and $X_2'$ to approximate the task.
3. OurCL-SUP: FACTORCL-SUP but removing the factorization so only two features $Z_1$ is optimized for both $I(X_1; X_2; Y)$ and $I(X_1; Y|X_2)$, $Z_2$ optimized for both $I(X_1; X_2; Y)$ and $I(X_2; Y|X_1)$.
4. OurCL-SSL: FACTORCL-SSL but also removing the factorization in the self-supervised setting.

The formulation of each ablation and implementation can be found in Appendix D.1.

## 4.1 Controlled experiments on synthetic datasets

**Synthetic data generation**: We begin by generating data with controllable ratios of task-relevant shared and unique information. Starting with a set of latent vectors $w_1, w_2, w_s \sim \mathcal{N}(0_d, \Sigma_d^2), d = 50$ representing information unique to $X_1, X_2$ and common to both respectively, the concatenated vector $[w_1, w_s]$ is transformed into high-dimensional $x_1$ using a fixed transformation $T_1$ and likewise $[w_2, w_s]$ to $x_2$ via $T_2$. The label $y$ is generated as a function (with nonlinearity and noise) of varying ratios of $w_s, w_1$, and $w_2$ to represent shared and unique task-relevant information.

**Results**: In Figure 1, we show our main result on synthetic data comparing FACTORCL with existing CL baselines. FACTORCL consistently maintains the best performance, whereas SimCLR [13] and SupCon [41] see performance drops as unique information increases. Cross+Self [33, 36, 44, 89] recovers in fully unique settings (x-axis= 1.0) but suffers at other ratios.

**Representation probing information**: We run a probing experiment to compute how well different contrastive representations capture shared and unique information. In Table 1, for the $Z_i$'s learned by each method, we approximately compute $I(Z_i; w_1)$, $I(Z_i; w_2)$, and $I(Z_i; w_s)$ with respect to ground truth generative variables $w_s, w_1$, and $w_2$. As expected, existing methods such as SimCLR capture smaller amounts of unique information (roughly 4 bits in $I(Z_i; w_1)$ and $I(Z_i; w_2)$), focusing instead on learning $I(Z_i; w_s)$ (12 bits). Cross+self captures slightly larger $I(Z_i; w_2) = 4.26$, and SupCon with labeled data captures up to 5 bits of unique information. Our FACTORCL approach captures 7 bits of unique information and maintains 10 bits of shared information, with total information captured higher than the other approaches. Furthermore, $\{Z_{S_1}, Z_{S_2}\}$ capture more information about $w_s$, $Z_{U_1}$ about $w_1$, and $Z_{U_2}$ about $w_2$, indicating that factorization in our approach is successful.

## 4.2 Self-supervised multimodal learning with low redundancy and high uniqueness

**Multimodal fusion datasets**: We use a large collection of real-world datasets provided in Multi-Bench [45], where we expect varying ratios of shared and unique information important for the task, to compare FACTORCL with other CL baselines:

1. MIMIC [38]: mortality and disease prediction from $36, 212$ medical records (tabular patient data and medical time-series sensors from ICU).
2. MOSEI [93]: multimodal sentiment and emotion benchmark with $23, 000$ monologue videos.
3. MOSI [91]: multimodal sentiment analysis from $2, 199$ YouTube videos.
4. UR-FUNNY [27]: a dataset of humor detection from more than $16, 000$ TED talk videos.
5. MUsTARD [12]: a corpus of 690 videos for research in sarcasm detection from TV shows.
6. IRFL [88]: $6, 697$ matching images and figurative captions (rather than literal captions).

Together, these datasets cover seven different modalities from the healthcare, affective computing, and multimedia research areas and total more than $84, 000$ data points. For MIMIC with tabular and medical sensor inputs, we train self-supervised CL models on top of raw modality inputs. For IRFL with image and caption inputs, we start with a pretrained CLIP model [61] and perform continued pre-training to update CLIP weights with our FACTORCL objectives, before linear classifier testing. For the remaining four video datasets, we train self-supervised CL models starting from standard pre-extracted text, video, and audio features [45]. Please refer to Appendix D.2 for experimental details. We release our code and models at `https://github.com/pliang279/FactorCL`.

Table 2: Results on MultiBench [45] datasets with varying shared and unique information: FACTORCL achieves strong results vs self-supervised (top 5 rows) and supervised (bottom 3 rows) baselines that do not have unique representations, factorization, upper-bounds to remove irrelevant information, and multimodal augmentations.

| Model | $(X_1; X_2)$ | $(X_i; X_i')$ | $(X_1; X_2\|Y)$ | $(X_2'')$ | Fact | MIMIC | MOSEI | MOSI | UR-FUNNY | MUSTARD |
|---|---|---|---|---|---|---|---|---|---|---|
| SimCLR [13] | ✓ | ✗ | ✗ | ✗ | ✗ | 66.67% | 71.03% | 46.21% | 50.09% | 53.48% |
| Cross+Self [81] | ✓ | ✓ | ✗ | ✗ | ✗ | 65.20% | 71.04% | 46.92% | 56.52% | 53.91% |
| Cross+Self+Fact [89] | ✓ | ✓ | ✗ | ✗ | ✓ | 65.49% | 71.07% | 52.37% | 59.91% | 53.91% |
| OurCL-SSL | ✓ | ✓ | ✓ | ✓ | ✗ | 65.22% | 71.16% | 48.98% | 58.79% | 53.98% |
| FACTORCL-SSL | ✓ | ✓ | ✓ | ✓ | ✓ | **67.34%** | **74.88%** | **52.91%** | **60.50%** | **55.80%** |
| SupCon [41] | ✗ | ✗ | ✓ | ✗ | ✗ | 67.37% | 72.71% | 47.23% | 50.98% | 52.75% |
| OurCL-SUP | ✓ | ✓ | ✓ | ✗ | ✗ | 68.16% | 71.15% | 65.32% | 58.32% | 65.05% |
| FACTORCL-SUP | ✓ | ✓ | ✓ | ✗ | ✓ | **76.79%** | **77.34%** | **70.69%** | **63.52%** | **69.86%** |

**Multimodal fusion results**: From Table 2, FACTORCL significantly outperforms the baselines that do not capture both shared and unique information in both supervised and self-supervised settings, particularly on MUSTARD (where unique information expresses sarcasm, such as sardonic facial expressions or ironic tone of voice), and on MIMIC (with unique health indicators and sensor readings). In Table 3, we also show that FACTORCL substantially improves the state-of-the-art in classifying images and figurative captions which are not literally descriptive of the image on IRFL, outperforming zero-shot and fine-tuned CLIP [61] as well as continued pre-training baselines on top of CLIP.

**Modeling ablations**: In Table 2, we also carefully ablate each component in our method and indicate either existing baselines or newly-run ablation models.

1. **Factorized representations**: In comparing FACTORCL-SSL with OurCL-SSL, and also FACTORCL-SUP with OurCL-SUP, we find that factorization is critical: without it, performance drops on average $6.1\%$, with performance drop as high as $8.6\%$ for MIMIC.
2. **Information removal via upper bound**: By comparing FACTORCL with SimCLR, Cross+Self, and Cross+Self+Fact, and SupCon that only seek to capture task-relevant information via contrastive lower bounds on MI, we find that separately modeling the task-relevant information (to be captured) and task-irrelevant information (to be removed) is helpful. Without removing task-irrelevant information via the upper-bound objective, performance drops on average $13.6\%$, with performance drops as high as $23.5\%$ for the MOSI dataset. We also found that training was more difficult without this objective, which is expected due to overwhelming superfluous information from the dataset [93].
3. **Multimodal augmentations**: Finally, we investigate the differences between separate unimodal augmentations (FACTORCL-IndAug in Table 3) versus a joint multimodal augmentation (FACTORCL-SSL) on the IRFL dataset. We choose this dataset since its images and captions are the easiest to visualize (see Figure 4 for augmentations from both strategies). In the self-supervised setting, we find that multimodal augmentations achieve $95\%$ performance, higher than the $92\%$ for separate unimodal augmentations, and both outperform baselines SimCLR and Cross+Self.

**Ablations on $S, U_1$ and $U_2$**: In Table 4, we also test FACTORCL when training linear classifiers on top of only shared $\{Z_{S_1}, Z_{S_2}\}$ and unique $Z_{U_1}$, $Z_{U_2}$ separately. We call these models FACTORCL-$S$, FACTORCL-$U_1$, and FACTORCL-$U_2$. Immediately, we observe that performance drops as compared to the full FACTORCL model, indicating that both shared and unique information are critical in real-world multimodal tasks. As expected, the best-performing submodel is the one that captures the region with the largest amount of task-relevant information: MOSEI and MOSI are known to include a lot of redundancy and unique information since language is very important for detecting sentiment [93], so FACTORCL-$S$ and FACTORCL-$U_2$ perform best. For sarcasm detection on MUS-TARD, video information is most important with FACTORCL-

Table 3: Continued pre-training on CLIP with our FACTORCL objectives on classifying images and figurative language.

| Task | IRFL |
|---|---|
| Zero-shot CLIP [61] | 89.15% |
| SimCLR [13] | 91.57% |
| Cross+Self [81, 89] | 95.18% |
| FACTORCL-IndAug | 92.77% |
| FACTORCL-SSL | **95.18%** |
| Fine-tuned CLIP [61] | 96.39% |
| SupCon [41] | 89.16% |
| FACTORCL-SUP | **98.80%** |

$U_1$ performing best ($59.4\%$), and ablation models are also the furthest away from full multimodal performance ($69.9\%$). This is aligned with intuition where sarcasm is expressed through tone of voice and visual gestures (high $U_1$), as well as from contradictions between language and video (higher multimodal performance).

Table 4: We ablate using only shared representations $\{Z_{S_1}, Z_{S_2}\}$, unique representation $Z_{U_1}$, and $Z_{U_2}$ separately for prediction. Both shared and unique information are critical in real-world multimodal tasks.

| Model | MIMIC | MOSEI | MOSI | UR-FUNNY | MUSTARD |
|---|---|---|---|---|---|
| FACTORCL-$S$ | 63.77% | 77.17% | 70.12% | 63.42% | 57.25% |
| FACTORCL-$U_1$ | 55.90% | 77.06% | 70.11% | 62.00% | 59.42% |
| FACTORCL-$U_2$ | 69.08% | 71.01% | 52.33% | 54.35% | 53.62% |
| FACTORCL-SUP | **76.79%** | **77.34%** | **70.69%** | **63.52%** | **69.86%** |

**Additional results**: In Appendix D.3, we also verify FACTORCL in settings with abundant shared information, where we expect to recover the same performance as standard CL [13, 58, 72].

## 5 Related Work

**Contrastive learning** is a successful self-supervised learning paradigm for computer vision [11, 13, 14, 25, 28, 58], natural language [24, 54, 56], speech [5, 58, 63], and multimodal tasks [1, 37, 61]. Its foundational underpinnings are inspired by work in multiview information theory [23, 41, 70, 72, 76] studying the shared information between two views and whether they are necessary or sufficient in predicting the label. Recently, Wang et al. [81] and Kahana and Hoshen [39] discuss the limitations of assuming multiview redundancy and propose autoencoder reconstruction or unimodal contrastive learning to retain unique information, which resembles the Cross+self baselines in our experiments. We refer the reader to Shwartz-Ziv and LeCun [67] for a comprehensive review on multiview and contrastive learning. Our work also relates to conditional contrastive learning [17, 51, 78, 87], where positive or negative pairs are supposed to sample from conditional distributions.

**Multimodal contrastive learning** aims to align related data from different modalities, typically provided as positive pairs. This could be done via optimizing a contrastive objective for inter-modality pairs [1, 2, 37, 61], or both intra- and inter-modality data pairs [33, 36, 42, 44, 89]. Our work also relates to factorized representation learning, which primarily studies how to capture modality-specific information primarily in each modality and multimodal information redundant in both modalities [32, 75]. Prior work has used disentangled latent variable models [8, 30, 32, 75], mixture-of-experts [66], or product-of-experts [83] layer to explain factors in multimodal data.

**Information theory** [18, 65] has been used to study several phenomena in multimodal learning, including co-learning [62, 92] and multi-view learning [34, 76]. Due to its theoretical importance, several lower and upper bounds have been proposed for practical estimation [58–60, 84]. We build on the CLUB upper bound [16] to create a more accurate and stable bound. Our characterizations of shared and unique information are also related to partial information decomposition [82], co-information [7, 80], interaction information [53], and cross-domain disentanglement [35] research.

## 6 Conclusion

This paper studied how standard CL methods suffer when task-relevant information lies in regions unique to each modality, which is extremely common in real-world applications such as sensor placement, medical testing, and multimodal interaction. In response, we proposed FACTORCL, a new method expanding CL techniques through the use of factorized representations, removing task-irrelevant information via upper bounds on MI, and multimodal data augmentations suitable for approximating the unobserved task. Based on FACTORCL's strong performance, there are several exciting directions in extending these ideas for masked and non-contrastive pre-training; we further discuss broader impacts and limitations of this line of work in Appendix A.

## Acknowledgements

This material is based upon work partially supported by Meta, National Science Foundation awards 1722822 and 1750439, and National Institutes of Health awards R01MH125740, R01MH132225, R01MH096951 and R21MH130767. PPL is supported in part by a Siebel Scholarship and a Waibel Presidential Fellowship. RS is supported in part by ONR grant N000142312368 and DARPA FA87502321015. One of the aims of this project is to understand the comfort zone of people for better privacy and integrity. Any opinions, findings, conclusions, or recommendations expressed in this material are those of the author(s) and do not necessarily reflect the views of the sponsors, and no official endorsement should be inferred. Finally, we would also like to acknowledge feedback from anonymous reviewers who significantly improved the paper and NVIDIA's GPU support.

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

# Appendix

## A Broader Impact

Multimodal data and self-supervised models are ubiquitous in a range of real-world applications. This paper is our attempt at broadening the applicability of self-supervised contrastive methods to a wider range of multimodal tasks beyond those that exhibit multi-view redundancy. We believe that special care must be taken to ensure that these models are safely deployed for real-world benefit:

**Time and space complexity**: Modern multimodal models are large and take up a significant amount of carbon footprint during training and testing. As compared to heuristic combinations of cross-modal and single-modality CL [33, 36, 44, 64, 81, 85, 89], we believe that FACTORCL does not significantly increase complexity: (1) upper bounds on MI can be estimated "for free" by directly plugging in the optimal critic from $I_{\text{NCE}}$, and (2) removal of task-irrelevant information via $I(X_1; X_2 | X_1', X_2')$ shares encoders with $I_{\text{NCE}}$, and (3) separate unimodal augmentations perform well enough in practice. We also release our code and models so that they can be evaluated quickly on new tasks, which can amortize complexity costs.

**Privacy and security**: There may be privacy risks associated with making predictions from multi-modal data of recorded human behaviors and medical data (i.e., the datasets used in our experiments for analysis of sentiment, emotions, personality, sarcasm, and humor, as well as disease prediction from medical data). We have followed best practices in maintaining the privacy and safety of these datasets: (1) the creators of these video datasets have taken the appropriate steps to only access public data that participants or content creators have consented for public release (creative commons license and following fair use guidelines of YouTube) [12, 27, 93], (2) MIMIC has been rigorously de-identified in accordance with Health Insurance Portability and Accountability Act (HIPAA) such that all possible personal information has been removed from the dataset [38], (3) all video data was also anonymized and stripped of all personal (e.g., personally identifiable information) and protected attributes (e.g., race, gender), (4) all models trained on affect recognition datasets use only pre-extracted non-invertible features that rely on general visual or audio features such as the presence of a smile or magnitude of voice which cannot be used to identify the speaker [91, 93], and (5) we studied the videos collected in these affective computing datasets and found no offensive words used or personal attacks recorded in the video. Finally, we only use these datasets for research purposes and emphasize that any multimodal models trained to perform prediction should only be used for scientific study and should not in any way be used for real-world harm.

**Social biases**: We acknowledge risks of social bias due to imbalanced datasets, resulting in potential biases surrounding gender, race, and ethnicity, among others [9, 46]. We note that our FACTORCL approach has a close link with conditional CL [51], which can also be adapted to condition on sensitive attributes and therefore reduce bias. Studying these research questions is an important direction for future work.

**Future work**: We discuss more limitations and potential future work in this direction. Firstly, optimizing our objectives using better MI lower and upper bounds such as in Guo et al. [26] and Sordoni et al. [69], could improve the performance for inputs of higher dimension and complex modality. Next, the current data augmentation method requires one to pick augmentations to approximately satisfy Definition 4; future work could extend InfoMin [72] to automatically generate data augmentations to satisfy Definition 4, or leverage future progress in multimodal generative models for data augmentation. Lastly, future work could quantify whether shared or unique information is more important for different tasks and reweight the terms in the FACTORCL objective to suit the tasks.

## B Analysis of Multi-view Contrastive Learning

**Multi-view shared information** describes the extent and dimensions in which information can be shared across different views. The presence of shared information is often in contrast to unique information that exists solely in a single modality, and can be formalized via information theory:

**Definition 5.** *(Shared information) Given $X_1$ and $X_2$, $I(X_1; X_2) = \int p(x_1, x_2) \log \frac{p(x_1, x_2)}{p(x_1) p(x_2)}$ measures the degree of information-theoretic shared information between $X_1$ and $X_2$.*

**Definition 6.** *(Task-relevant shared information) Given $X_1$, $X_2$, and a target $Y$, $I(X_1; X_2; Y) = I(X_1; X_2) - I(X_1; X_2 | Y) = \int p(x_1, x_2) \log \frac{p(x_1, x_2)}{p(x_1) p(x_2)} - \int p(x_1, x_2 | y) \log \frac{p(x_1, x_2 | y)}{p(x_1 | y) p(x_2 | y)}$ measures*

*the amount of task-relevant shared information between $X_1$ and $X_2$ for predicting $Y$. $I(X_1; X_2|Y)$ represents the task-irrelevant shared information.*

**Learning shared information via contrastive learning**: Current approaches for multi-view contrastive learning model shared information $I(X_1; X_2)$ (and subsequently task-relevant shared information $I(X_1; X_2; Y)$ during downstream task fine-tuning), without modeling unique information.

$$Z_1 = \underset{Z_1 := f_\theta(X_1)}{\arg\max} \ I(Z_1; X_2), Z_2 = \underset{Z_2 := f_\theta(X_2)}{\arg\max} \ I(X_1; Z_2). \tag{19}$$

Optimizing for $I(X_1; X_2)$ is performed via a surrogate loss during self-supervised pre-training (where we do not have access to the label $Y$) by maximizing the InfoNCE objective:

$$\text{INFONCE} = \sup_f \mathbb{E}_{\substack{x_1, x_2^+ \sim p(x_1, x_2) \\ x_2^- \sim p(x_2)}} \left[ \log \frac{\exp f(x_1, x_2^+)}{\sum_k \exp f(x_1, x_2^-)} \right], \tag{20}$$

Oord et al. [58] show that $I(X_1; X_2) \geq \log k - \mathcal{L}_{\text{NCE}}(X_1; X_2)$ where $\mathcal{L}_{\text{NCE}}(X_1; X_2)$ is negative of INFONCE and is the loss to minimize (rather than maximize) in training. NCE falls into a broader class of contrastive learning methods [13, 15, 28, 41, 61] that model the ratio between joint densities $p(x_1, x_2)$ and product of marginal densities $p(x_1)p(x_2)$ using positive and negative samples [57, 59, 60, 79, 84] or probabilistic classifiers [55, 77], all of which can also be used to capture shared information.

Tian et al. [71] argues that the optimal view of contrastive learning is also minimal: the minimal representations only extract relevant information of the contrastive task (maximizing the shared part) and throw away other information. Therefore, from this minimal assumption, we have $I(Z_1; Y|X_2) = 0$ and $I(Z_2; Y|X_1) = 0$ as minimal $Z_1$ and $Z_2$ only captures task-relevant information from the shared part. By conditioning on $X_1$ or $X_2$, the shared part is removed, and $Z_1$ and $Y$ (or $Z_2$ and $Y$) do not share information.

Lastly, we restate the multi-view non-redundancy from Definition 2:

**Definition 7.** *(Multi-view non-redundancy)* $\exists \epsilon > 0$ *such that* $I(X_1; Y|X_2) > \epsilon$ *or* $I(X_2; Y|X_1) > \epsilon$.

We would like to compare and clarify the differences between the multiview redundancy assumption in Eq.(1) and the multi-view nonredundancy in Def. 7. The multiview redundancy assumption in Eq.(1) states that the task-relevant information from the unique part is minimal ($\leq \epsilon$). The multiview non-redundancy states the opposite: the task-relevant information from the unique part is nonzero and nonminimal, as it is not bounded by $\epsilon$. Next we briefly clarify the relationship between these two assumptions and the InfoMin assumption: $I(Z_1; Y|X_2) = I(Z_2; Y|X_1) = 0$. InfoMin is about representation $Z$ while the redundancy assumptions are only about data $X$. InfoMin states that the optimal (sufficient and minimal) representation learns task-relevant information only from the shared part, as we discussed in the paragraph above. We empirically checked the two assumptions: Tables 1 and 4 in the main text show that the multiview non-redundancy assumption holds empirically, and Table 11 shows that the InfoMin assumption holds empirically.

We now show the limitations of CL methods, first restating the Theorem here:

**Theorem 4.** *(Suboptimality of standard CL) When there is multi-view non-redundancy as in Definition 7, given optimal representations* $\{Z_1, Z_2\}$ *that satisfy Eq.(19 and* $I(Z_1; Y|X_2) = I(Z_2; Y|X_1) = 0$ *[72], we have that*

$$I(Z_1, Z_2; Y) = I(X_1, X_2; Y) - I(X_1; Y|X_2) - I(X_2; Y|X_1) = I(X_1; X_2) - I(X_1; X_2|Y) < I(X_1, X_2; Y). \tag{21}$$

*Proof.* Since $Z_1$ and $Z_2$ maximize $I(X_1; X_2)$ we have that $I(Z_1; X_2) = I(X_1; Z_2) = I(X_1; X_2)$ so $I(Z_1; X_2; Y) = I(X_1; Z_2; Y) = I(X_1; X_2; Y)$ and $I(Z_1; X_2|Y) = I(X_1; Z_2|Y) = I(X_1; X_2|Y)$.

We now show the relationship between $I(X_1, X_2; Y)$, which is the total information $X_1, X_2$ contributes towards predicting $Y$ in classic supervised learning, with $I(Z_1, Z_2; Y)$, which is the information that our learned self-supervised representations can contribute towards $Y$ during supervised fine-tuning. We first derive the relationship between $I(Z_1; Y)$ and $I(X_1; Y)$:

$$I(Z_1; Y) = I(Z_1; X_2; Y) + I(Z_1; Y|X_2) \tag{22}$$
$$= I(X_1; X_2; Y) + I(Z_1; Y|X_2) \tag{23}$$
$$= I(X_1; Y) - I(X_1; Y|X_2) + I(Z_1; Y|X_2) \tag{24}$$
$$= I(X_1; Y) - I(X_1; Y|X_2) \tag{25}$$

Given $X_1$, we further derive a relationship between $I(Z_2; Y|Z_1)$ and $I(X_2; Y|X_1)$:

$$I(Z_2; Y|Z_1) = I(Z_2; X_1; Y|Z_1) + I(Z_2; Y|Z_1, X_1) \tag{26}$$

$$= I(X_1; X_2; Y|Z_1) + I(Z_2; Y|Z_1, X_1) \tag{27}$$

$$= I(X_1; X_2; Y|Z_1) + I(Z_2; Y|X_1) \tag{28}$$

$$= I(X_2; Y|Z_1) - I(X_2; Y|X_1, Z_1) + I(Z_2; Y|X_1) \tag{29}$$

$$= I(X_2; Y|Z_1) - I(X_2; Y|X_1) + I(Z_2; Y|X_1) \tag{30}$$

$$= I(X_2; Y) - I(Z_1; X_2; Y) - I(X_2; Y|X_1) + I(Z_2; Y|X_1) \tag{31}$$

$$= I(X_2; Y) - I(X_1; X_2; Y) - I(X_2; Y|X_1) + I(Z_2; Y|X_1) \tag{32}$$

$$= I(X_2; Y|X_1) - I(X_2; Y|X_1) + I(Z_2; Y|X_1) = 0 \tag{33}$$

In Eqs.(28) and (30), we use the fact that conditioning on $Z_1$ and $X_1$ jointly reduces to conditioning on $X_1$ since $Z_1$ is deterministically obtained from $X_1$, and in Eq.(32) we use the definition of learning $Z_s$ to maximize $I(X_1; X_2)$ so $I(Z_1; X_2; Y) = I(X_1; X_2; Y)$. Finally, adding both terms up,

$$I(Z_1, Z_2; Y) = I(Z_1; Y) + I(Z_2; Y|Z_1) \tag{34}$$

$$= I(X_1; Y) - I(X_1; Y|X_2) \tag{35}$$

$$= I(X_1; X_2; Y) \tag{36}$$

$$= I(X_1, X_2; Y) - I(X_1; Y|X_2) - I(X_2; Y|X_1) \tag{37}$$

$$= I(X_1; X_2) - I(X_1; X_2|Y) \tag{38}$$

gives the desired result. $\square$

**Bayes error rate.** The Bayes error rate $P_e(Z_1, Z_2) := 1 - \mathbb{E}_{P_{Z_1, Z_2}}\left[\max_{y \in Y} P\left(\hat{Y} = y \mid z_1, z_2\right)\right]$ of contrastive representations $\{Z_1, Z_2\}$ is given by:

$$P_e \le 1 - \exp\left[I(X_1, X_2; Y) - I(X_1; Y|X_2) - I(X_2; Y|X_1) - H(Y)\right] \tag{39}$$

$$= 1 - \exp\left[I(X_1; X_2; Y) - H(Y)\right] \tag{40}$$

*Proof.* We use the inequality between $P_e$ and $H(Y|Z)$ [22, 76, 81]:

$$-\ln(1 - P_e) \le H(Y|Z), \text{ or equivalently, } P_e \le 1 - \exp[-H(Y|Z)] \tag{41}$$

If we regard $Z$ as the joint of $Z_1$ and $Z_2$, then we have

$$P_e \le 1 - \exp[-H(Y|Z_1, Z_2)] \tag{42}$$

We further expand $H(Y|Z_1, Z_2)$ by definition of mutual information, $I(X; Y) = H(X) - H(X|Y)$, Theorem 4, and the $I(X_1; X_2; Y) = I(X_1; X_2) - I(X_1; X_2|Y)$:

$$H(Y|Z_1, Z_2) = H(Y) - I(Z_1, Z_2; Y) \tag{43}$$

$$= H(Y) - I(X_1, X_2; Y) + I(X_1; Y|X_2) + I(X_2; Y|X_1) \tag{44}$$

$$= H(Y) - I(X_1; X_2) + I(X_1; X_2|Y) \tag{45}$$

$$= H(Y) - I(X_1; X_2; Y) \tag{46}$$

Plugging in Eq.(42), we have

$$P_e \le 1 - \exp[-H(Y|Z_1, Z_2)] \tag{47}$$

$$= 1 - \exp[-(H(Y) - I(X_1, X_2; Y) + I(X_1; Y|X_2) + I(X_2; Y|X_1))] \tag{48}$$

$$= 1 - \exp[-H(Y) + I(X_1, X_2; Y) - I(X_1; Y|X_2) - I(X_2; Y|X_1)] \tag{49}$$

and

$$P_e \le 1 - \exp[-H(Y|Z_1, Z_2)] \tag{50}$$

$$= 1 - \exp[-(H(Y) - I(X_1; X_2; Y))] \tag{51}$$

$$= 1 - \exp[-H(Y) + I(X_1; X_2; Y)] \tag{52}$$

resulting in the Bayes error rate as desired. $\square$

## C FACTORIZED CONTRASTIVE LEARNING

### C.1 Contrastive estimators

**Theorem 5.** *(Contrastive estimators for $I(X_1; X_2)$) Defining the NCE estimator and NCE-CLUB estimator as follows,*

$$I_{\text{NCE}}(X_1; X_2) = \mathbb{E}_{\substack{x_1, x_2^+ \sim p(x_1, x_2) \\ x_2^- \sim p(x_2)}} \left[ \log \frac{\exp f(x_1, x_2^+)}{\sum_k \exp f(x_1, x_2^-)} \right] \tag{53}$$

$$I_{\text{NCE-CLUB}}(X_1; X_2) = \mathbb{E}_{x_1, x_2^+ \sim p(x_1, x_2)} \left[ f^*(x_1, x_2^+) \right] - \mathbb{E}_{\substack{x_1 \sim p(x_1) \\ x_2^- \sim p(x_2)}} \left[ f^*(x_1, x_2^-) \right] \tag{54}$$

*where $f^*(x_1, x_2)$ is the optimal critic from $I_{\text{NCE}}$ plugged into the $I_{\text{CLUB}}$ objective [16]. We call the proposed plug-in objective Eq.(11) $I_{\text{NCE-CLUB}}$, and obtain lower and upper bounds on $I(X_1; X_2)$:*

$$I_{\text{NCE}}(X_1; X_2) \le I(X_1; X_2) \le I_{\text{NCE-CLUB}}(X_1; X_2). \tag{55}$$

*Proof.* The lower bound $I_{\text{NCE}}(X_1; X_2) \le I(X_1; X_2)$ follows from Oord et al. [58]: optimizing the objective leads to an optimal critic $f^* = \log p(x_2|x_1) + c(x_2)$ [60] or without loss of generality $f^* = \log p(x_1|x_2) + c(x_1)$, where $c(\cdot)$ is an arbitrary deterministic function. Plugging the optimal critic into the $I_{\text{NCE}}(X_1; X_2)$ gives the result: $I_{\text{NCE}}(X_1; X_2) + \log N \le I(X_1; X_2)$ [58, 60].

Next, the original $I_{\text{CLUB}}(X_1; X_2)$ [16] is defined as:

$$I_{\text{CLUB}}(X_1; X_2) \coloneqq \mathbb{E}_{p(x_1, x_2)} \left[ \log p(x_2|x_1) \right] - \mathbb{E}_{p(x_1)p(x_2)} \left[ \log p(x_2|x_1) \right]. \tag{56}$$

As mutual information is symmetric w.r.t $x_1$ and $x_2$: $I(X_1; X_2) = I(X_2; X_1)$, without loss of generality, we have:

$$I_{\text{CLUB}}(X_1; X_2) = I_{\text{CLUB}}(X_2; X_1) = \mathbb{E}_{p(x_1, x_2)} \left[ \log p(x_1|x_2) \right] - \mathbb{E}_{p(x_1)p(x_2)} \left[ \log p(x_1|x_2) \right] \tag{57}$$

The formulation above assumes $p(x_1|x_2)$ is known, which is satisfied when we use the optimal critic $f^* = \log p(x_1|x_2) + c(x_1)$ from $I_{\text{NCE}}(X_1; X_2)$. Plugging the optimal critic $f^*$ into $I_{\text{CLUB}}(X_1; X_2)$, we obtain a desired upper bound $I_{\text{NCE-CLUB}}(X_1; X_2)$ of $I(X_1; X_2)$:

$$I_{\text{NCE-CLUB}}(X_1; X_2) = \mathbb{E}_{p(x_1, x_2)} \left[ \log p(x_1|x_2) + c(x_1) \right] - \mathbb{E}_{p(x_1)p(x_2)} \left[ \log p(x_1|x_2) + c(x_1) \right] \tag{58}$$

$$= \mathbb{E}_{p(x_1, x_2)} \left[ \log p(x_1|x_2) \right] + \mathbb{E}_{p(x_1, x_2)} \left[ c(x_1) \right] - \mathbb{E}_{p(x_1)p(x_2)} \left[ \log p(x_1|x_2) \right] - \mathbb{E}_{p(x_1)p(x_2)} \left[ c(x_1) \right] \tag{59}$$

$$= \mathbb{E}_{p(x_1, x_2)} \left[ \log p(x_1|x_2) \right] - \mathbb{E}_{p(x_1)p(x_2)} \left[ \log p(x_1|x_2) \right] \tag{60}$$

$$\ge I(X_1; X_2). \tag{61}$$

Eq.(59) is from the linearity of expectation, Eq.(60) is from the fact that $c(x_1)$ is not a function of $x_2$ and therefore $\mathbb{E}_{p(x_1, x_2)} \left[ c(x_1) \right] = \mathbb{E}_{p(x_1)p(x_2)} \left[ c(x_1) \right] = \mathbb{E}_{p(x_1)} \left[ c(x_1) \right]$, and Eq.(61) is directly from the original $I_{\text{CLUB}}(X_1; X_2)$ paper [16]. □

### C.2 Unimodal and multimodal augmentations

We first restate the definitions of optimal single-view and multi-view augmentation:

**Definition 8.** *(Optimal unimodal augmentation) $X_1'$ is an optimal unimodal augmentation for $X_1$ when $I(X; X') = I(X; Y)$, which implies that the only information shared between $X$ and $X'$ is task-relevant with no irrelevant noise.*

**Definition 9.** *(Optimal multimodal augmentation) $X_1'$ and $X_2'$ are optimal multimodal augmentation for $X_1$ and $X_2$ when $I(X_1, X_2; X_1', X_2') = I(X_1, X_2; Y)$, which implies that the only information shared between $X_1, X_2$ and $X_1', X_2'$ is task-relevant with no irrelevant noise.*

When are these assumptions satisfied? $I(X; X') = I(X; Y)$ holds when all information shared between $X$ and $X'$ is task-relevant, which implies that the augmentation keeps task-relevant information constant while changing task-irrelevant information. In the case of image classification, task-relevant information is the object in the picture, while task-irrelevant information is the background. To satisfy $I(X_1, X_2; X_1', X_2') = I(X_1, X_2; Y)$, by the chain rule of MI, we augment in two steps:

$$\text{Unimodal aug: } X_1' \text{ s.t. } I(X_1; X_1') = I(X_1; Y), \tag{62}$$

$$\text{Unique aug: } X_2' \text{ s.t. } I(X_2; X_2'|X_1) = I(X_2; Y|X_1). \tag{63}$$

Table 5: More examples of optimal single-view and multi-view augmentations.

| Dataset | $X_1$ | $X_2$ | $X_1'$ | Standard Aug $X_2'$ | **Unique Aug** $X_2''$ |
|---|---|---|---|---|---|
| Cartoon | Caption | Image | Word Masking | Crop + Flip + Resize | Flip + Resize |
| MIMIC | Signals | Tables | Time Warping | CutMix [90] on All Features | CutMix on Nonclinical Features |
| MOSEI | Transcripts | Video+Audio | Word Masking | Noise Injection on Any Frames | Noise Injection on Silent Frames |
| UR-FUNNY | Transcripts | Video+Audio | Word Masking | Noise Injection on Any Frames | Noise Injection on Silent Frames |
| MUsTARD | Transcripts | Video+Audio | Word Masking | Noise Injection on Any Frames | Noise Injection on Silent Frames |

the second step is the *unique augmentation*: after observing $X_1$, we create augmented $X_2'$ from $X_2$ to keep the task-relevant information but meanwhile do not affect any information from $X_1$. In Table 5, we include some more examples of how unique augmentations could be designed across different datasets.

**Final objectives**: If Definitions 8 and 9 are both satisfied, we can substitute contrastive estimators in the following equations:

$$I_{\text{NCE}}(X_1; X_2|Y) = \mathbb{E}_{p(y)}\left[\mathbb{E}_{\substack{x_1,x_2^+\sim p(x_1,x_2|y)\\x_2^-\sim p(x_2|y)}}\left[\log\frac{\exp f(x_1,x_2^+,y)}{\sum_k \exp f(x_1,x_2^-,y)}\right]\right] \tag{64}$$

$$I_{\text{NCE-CLUB}}(X_1; X_2|Y) = \mathbb{E}_{p(y)}\left[\mathbb{E}_{x_1,x_2^+\sim p(x_1,x_2|y)}\left[f^*(x_1,x_2^+,y)\right] - \mathbb{E}_{\substack{x_1\sim p(x_1|y)\\x_2^-\sim p(x_2|y)}}\left[f^*(x_1,x_2^-,y)\right]\right] \tag{65}$$

by replacing $I(X_i;Y)$ terms with $I(X_i;X_i')$ and replacing $I(X_1;X_2|Y)$ terms with $I(X_1;X_2|X_1',X_2')$:

$$I_{\text{NCE}}(X_1; X_2|X_1',X_2') = \mathbb{E}_{p(x_1',x_2')}\left[\mathbb{E}_{\substack{x_1,x_2^+\sim p(x_1,x_2|x_1',x_2')\\x_2^-\sim p(x_2|x_1',x_2')}}\left[\log\frac{\exp f(x_1,x_2^+,x_1',x_2')}{\sum_k \exp f(x_1,x_2^-,x_1',x_2')}\right]\right] \tag{66}$$

$$I_{\text{NCE-CLUB}}(X_1; X_2|X_1',X_2') = \mathbb{E}_{p(x_1',x_2')}\Big[\mathbb{E}_{x_1,x_2^+\sim p(x_1,x_2|x_1',x_2')}[f^*(x_1,x_2^+,x_1',x_2')]$$
$$- \mathbb{E}_{\substack{x_1\sim p(x_1|x_1',x_2')\\x_2^-\sim p(x_2|x_1',x_2')}}[f^*(x_1,x_2^-,x_1',x_2')]\Big] \tag{67}$$

### C.2.1 Implementing conditional CL via kernel

We restate our objectives below:

$$I_{\text{NCE}}(X_1; X_2|X_1',X_2') = \mathbb{E}_{p(x_1',x_2')}\left[\mathbb{E}_{\substack{x_1,x_2^+\sim p(x_1,x_2|x_1',x_2')\\x_2^-\sim p(x_2|x_1',x_2')}}\left[\log\frac{\exp f(x_1,x_2^+,x_1',x_2')}{\sum_k \exp f(x_1,x_2^-,x_1',x_2')}\right]\right] \tag{68}$$

$$I_{\text{NCE-CLUB}}(X_1; X_2|X_1',X_2') = \mathbb{E}_{p(x_1',x_2')}\Big[\mathbb{E}_{x_1,x_2^+\sim p(x_1,x_2|x_1',x_2')}[f^*(x_1,x_2^+,x_1',x_2')]$$
$$- \mathbb{E}_{\substack{x_1\sim p(x_1|x_1',x_2')\\x_2^-\sim p(x_2|x_1',x_2')}}[f^*(x_1,x_2^-,x_1',x_2')]\Big] \tag{69}$$

However, sampling from $p(\cdot|x_1',x_2')$ is hard. Since $X_1', X_2'$ are continuous variables, directly sampling from the conditional distributions $p(\cdot|x_1',x_2')$ may be difficult; training a generative model $p_\theta(x_1,x_2|x_1',x_2')$ from augmented data $x_1',x_2'$ to original data $x_1,x_2$ can be expensive and nontrivial in a multimodal setup. In this work, we implement the conditioning in $p(x_1,x_2|x_1',x_2')$ through concatenation and the details are in Appendix D.1. Here we discuss an alternative solution to this problem introduced by Tsai et al. [78]. It leverages kernel methods for conditional sampling in contrastive learning by assigning weights to each sampled data given the kernel similarity between conditioned variables, avoiding directly sampling from the conditional distributions or training generative models. In our formulation, given multimodal input $(x_1,x_2)$ with their augmentation $(x_1',x_2')$, we can simply use the technique from [78] to estimate $I_{\text{NCE}}(X_1; X_2|X_1',X_2')$ and $I_{\text{NCE-CLUB}}(X_1; X_2|X_1',X_2')$, where the kernel measures the similarity between different pairs $(x_1',x_2')$ of the conditional variable

$X_1, X_2$. Specifically,

$$I_{\text{NCE}}(X_1; X_2 | X_1', X_2') = \mathbb{E}_{p(x_1, x_2, x_1', x_2')} \left[ \log \frac{\exp f(x_1, x_2^+)}{\exp f(x_1, x_2) + n * \left[ K_{X_1 \perp\!\!\!\perp X_2 | X_1', X_2'} \right]_{ii}} \right] \quad (70)$$

$$I_{\text{NCE-CLUB}}(X_1; X_2 | X_1', X_2') = \mathbb{E}_{p(x_1, x_2, x_1', x_2')} \left[ f^*(x_1, x_2) - \log \left[ K_{X_1 \perp\!\!\!\perp X_2 | X_1', X_2'} \right]_{ii} \right] \quad (71)$$

where $K_{X_1 \perp\!\!\!\perp X_2 | X_1', X_2'} = K_{X_1 X_2} (K_{X_1' X_2'} + \lambda \mathbf{I})^{-1} K_{X_1' X_2'}$ and $\left[ K_{X_1 \perp\!\!\!\perp X_2 | X_1', X_2'} \right]_{ii}$ is the $i$th row and $i$th column of $K_{X_1 \perp\!\!\!\perp X_2 | X_1', X_2'}$. $K_{X_1 X_2}$ is a kernel similarity matrix between $X_1$ and $X_2$, and $K_{X_1' X_2'}$ is a separate kernel similarity matrix between $X_1'$ and $X_2'$. $f^*$ is the optimal solution of Eq.(70). By leveraging the similarity $K_{X_1' X_2'}$ between conditional variables $X_1'$ and $X_2'$, $K_{X_1 \perp\!\!\!\perp X_2 | X_1', X_2'}$ transforms the similarity scores between $X_1$ and $X_2$ under unconditional distributions into similarity scores under conditional distributions. Note that the expectations in Eqs.(70) and (71) are taken over the joint distribution $p(x_1, x_2, x_1', x_2')$, which comes naturally after augmenting both modalities $X_1$ and $X_2$. This method could effectively alleviate the problem of sampling from conditional distributions in our formulation. We refer the reader to Tsai et al. [78] for more details.

## C.3 Final estimators in FACTORCL

**Theorem 6.** *(Contrastive estimators for shared and unique information). Under assumptions on single-view augmentations $I(X_1; Y) = I(X_1, X_1')$ (Definition 8) and optimal multi-view augmentation $X_2'$ such that $I(X_1, X_2; X_1', X_2') = I(X_1, X_2; Y)$ (Definition 9), we can define contrastive objectives for task-relevant shared and unique information with:*

$$S = I(X_1; X_2; Y) \geq I_{\text{NCE}}(X_1; X_2) - I_{\text{NCE-CLUB}}(X_1; X_2 | X_1', X_2') \quad (72)$$

$$U_i = I(X_i; Y | X_{-i}) \geq I_{\text{NCE}}(X_i; X_i') - I_{\text{NCE-CLUB}}(X_1; X_2) + I_{\text{NCE}}(X_1; X_2 | X_1', X_2') \quad (73)$$

*Proof.* The objectives follow from the fact that $I_{\text{NCE}}(X_1; X_2)$ and $I_{\text{NCE}}(X_1; X_2 | X_1', X_2')$ are lower bounds of $I(X_1; X_2)$ and $I(X_1; X_2 | Y)$ respectively, and $I_{\text{NCE-CLUB}}(X_1; X_2)$ and $I_{\text{NCE-CLUB}}(X_1; X_2 | X_1', X_2')$ are upper bounds of $I(X_1; X_2)$ and $I(X_1; X_2 | Y)$ respectively:

$$S = I(X_1; X_2; Y) = I(X_1; X_2) - I(X_1; X_2 | Y) \quad (74)$$

$$\geq I_{\text{NCE}}(X_1; X_2) - I_{\text{NCE-CLUB}}(X_1; X_2 | X_1', X_2') \quad (75)$$

$$U_i = I(X_i; Y | X_{-i}) = I(X_i; Y) - (I(X_1; X_2) - I(X_1; X_2 | Y)) \quad (76)$$

$$\geq I_{\text{NCE}}(X_i; X_i') - (I_{\text{NCE-CLUB}}(X_1; X_2) - I_{\text{NCE}}(X_1; X_2 | X_1', X_2')) \quad (77)$$

and symmetrically for $U_2$. $\qquad\square$

Now we show that FACTORCL learns both shared and unique task-relevant information. First, we restate the definition of the factorized representations:

$$Z_{S_1} = \underset{Z_1 = f_\theta(X_1)}{\arg\max} \, I(Z_1; X_2; Y), \qquad Z_{S_2} = \underset{Z_2 = f_\theta(X_2)}{\arg\max} \, I(Z_2; X_1; Y), \quad (78)$$

$$Z_{U_1} = \underset{Z_1 = f_\theta(X_1)}{\arg\max} \, I(Z_1; Y | X_2), \qquad Z_{U_2} = \underset{Z_2 = f_\theta(X_2)}{\arg\max} \, I(Z_2; Y | X_1). \quad (79)$$

where $I(Z_1; X_2; Y) = I(Z_1; X_2) - I(Z_1; X_2 | Y)$ is the shared information and $I(Z_2; X_1; Y) = I(Z_2; X_2) - I(Z_2; X_1 | Y)$ is the unique information.

**Theorem 7.** *(Optimality of FACTORCL) If $Z_{S_1}, Z_{S_2}, Z_{U_1}, Z_{U_2}$ perfectly maximize Eqs.(78-79) and the estimations in Eqs.(13-67) are tight, we obtain $I(X_1, X_2; Y) = I(Z_{S_1}; Z_{S_2}; Y) + I(Z_{U_1}; Y | Z_{S_2}) + I(Z_{U_2}; Y | Z_{S_1})$, suggesting that FACTORCL learns both shared and unique task-relevant information.*

*Proof.* Because $I(X_1, X_2; Y) = I(X_1; X_2; Y) + I(X_1; Y | X_2) + I(X_2; Y | X_1)$, it is sufficient to show that $I(X_1; X_2; Y) = I(Z_{S_1}; Z_{S_2}; Y), I(X_1; Y | X_2) = I(Z_{U_1}; Y | Z_{S_2})$ and $I(X_2; Y | X_1) = I(Z_{U_2}; Y | Z_{S_1})$.

First we show $I(X_1; X_2; Y) = I(Z_{S_1}; Z_{S_2}; Y)$. Crucially, by definition of how $Z_{S_1}$ and $Z_{S_2}$ are optimized to maximize $I(X_1; X_2; Y)$, we have that:

$$I(X_1; X_2; Y) = I(Z_{S_1}; X_2; Y) = I(Z_{S_2}; X_1; Y). \tag{80}$$

We can then obtain

$$I(X_1; X_2; Y) = I(X_1; Z_{S_2}; Y) \tag{81}$$
$$= I(X_1; Z_{S_2}; Y | Z_{S_1}) + I(Z_{S_1}; Z_{S_2}; X_1; Y) \tag{82}$$
$$= I(Z_{S_2}; Y | Z_{S_1}) - I(Z_{S_2}; Y | Z_{S_1}, X_1) + I(Z_{S_1}; Z_{S_2}; X_1; Y) \tag{83}$$
$$= I(Z_{S_2}; Y | Z_{S_1}) - I(Z_{S_2}; Y | X_1) + I(Z_{S_1}; Z_{S_2}; X_1; Y) \tag{84}$$
$$= I(Z_{S_2}; Y | Z_{S_1}) - I(Z_{S_2}; Y | X_1) + I(Z_{S_1}; Z_{S_2}; Y) \tag{85}$$
$$= I(Z_{S_2}; Y | Z_{S_1}) + I(Z_{S_1}; Z_{S_2}; Y) \tag{86}$$
$$= I(Z_{S_1}; Z_{S_2}; Y) \tag{87}$$

where Eq.(84) is because $Z_{S_1}$ are deterministically obtained from $S_1$ and Eq.(85) is because $Z_{S_1}$ maximizes the shared information. Finally, we go to Eq.(87) $I(Z_{S_2}; Y | Z_{S_1}) = 0$ as shown in Eqs.(26-33) using the fact that $Z_{S_1}$ is learned to maximize $I(X_1; X_2; Y)$ and $I(Z_{S_1}; X_2; Y) = I(X_1; Z_{S_2}; Y)$.

Next, we show $I(X_1; Y | X_2) = I(Z_{U_1}; Y | Z_{S_2})$:

$$I(Z_{U_1}; Y | Z_{S_2}) = I(Z_{U_1}; Y | Z_{S_2}, Z_{U_2}) + I(Z_{U_1}; Y; Z_{U_2} | Z_{S_2}), \tag{88}$$

which is by the chain rule of conditional mutual information. Then we show $I(Z_{U_1}; Y; Z_{U_2} | Z_{S_2}) = 0$:

$$I(Z_{U_1}; Y; Z_{U_2} | Z_{S_2}) = I(Z_{U_1}; Z_{U_2} | Z_{S_2}) - I(Z_{U_1}; Z_{U_2} | Y; Z_{S_2}) = 0 - 0 = 0 \tag{89}$$

This is because Eq.(79) leads to $I(Z_{U_1}; Y | X_2) = I(X_1; Y | X_2)$ and $I(Z_{U_2}; Y | X_1) = I(X_2; Y | X_1)$. If the estimations in Eqs.(13-67) are tight, by conditioning and by the previously stated $I(Z_{U_1}; Y | X_2) = I(X_1; Y | X_2)$, $Z_{U_1}$ tightly captures information from only $X_1$ and not in $X_2$. The same applies to $Z_{U_2}$. We have $I(Z_{U_1}; X_2) = I(Z_{U_2}; X_1) = I(Z_{U_1}; Z_{U_2}) = I(Z_{U_1}; Z_{U_2} | T) = 0$ with $T$ being an arbitrary random variable because no shared information exists between $Z_{U_1}$ and $Z_{U_2}$. Then we obtain:

$$I(Z_{U_1}; Y | Z_{S_2}, Z_{U_2}) = I(Z_{U_1}; Y | Z_{S_2}, Z_{U_2}, X_2) + I(Z_{U_1}; Y; X_2 | Z_{S_2}, Z_{U_2}) \tag{90}$$
$$= I(Z_{U_1}; Y | X_2) \tag{91}$$

We use the fact that conditioning on $Z_{S_2}, Z_{U_2}$ and $X_2$ jointly reduces to conditioning on $X_2$ since $Z_{S_2}$ and $Z_{U_2}$ are deterministically obtained from $X_2$. Lastly, since Eqs.(78-79) are satisfied, $Z_{U_1} = \arg\max_{Z_1 = f_\theta(X_1)} I(Z_1; Y | X_2)$ therefore $I(Z_{U_1}; Y | X_2) = I(X_1; Y | X_2)$. We have:

$$I(Z_{U_1}; Y | Z_{S_2}) = I(Z_{U_1}; Y | X_2) = I(X_1; Y | X_2). \tag{92}$$

The proof for $I(X_2; Y | X_1) = I(Z_{U_2}; Y | Z_{S_1})$ is similar. We now have shown that $I(X_1; X_2; Y) = I(Z_{S_1}; Z_{S_2}; Y)$, $I(X_1; Y | X_2) = I(Z_{U_1}; Y | Z_{S_2})$ and $I(X_2; Y | X_1) = I(Z_{U_2}; Y | Z_{S_1})$, adding up all LHS and RHS we have the theorem. $\square$

### C.4 Extensions to masking and non-contrastive learning

We now show how similar ideas can be extended to other popular self-supervised learning objectives, such as non-contrastive learning [6, 94] and masked pre-training [20, 29]. Importantly, this paper provides a new principle for multimodel self-supervised learning: (1) learning task-relevant information and (2) removing task-irrelevant information from both shared and unique parts across modalities. Our paper focuses on realizing this principle via multi-view information theory and contrastive learning. Below we provide two potential alternatives to realize this principle on non-contrastive and masking methods, respectively:

**Non-contrastive learning:** Methods such as Barlow Twins [94] and VICReg [6] use invariance and covariance regularizations to maximally preserve shared information in the embeddings across two modalities. However, the embeddings learned may contain only contain task-relevant information from the shared part and not unique parts. To use the principle in this paper to capture more task-relevant information from unique parts, one should perform VIC-regularization on $X_1$ features, on

$X_2$ features, and on $X_1, X_2$ cross-modal features. When performing VICReg on unimodal features, one should condition on the other modality when performing augmentation. Specifically, similar to the idea of multimodal augmentation in this paper, the augmentation of the second modality should not interfere with the shared part (i.e., do not augment regions referred to by the first modality), making the invariance and covariance regularization of the second modality focus on the augmented modality-unique features. This makes the model learn unique modality features that are not captured by the joint embedding from standard independent augmentations.

**Masking:** Conceptually, masking [20, 29] can be interpreted as leveraging unmasked regions in the same modality to predict masked regions or leveraging the other modality to predict the masked region. However, the learned representation may not be all task-relevant. To use the principle in this paper to exclude task-irrelevant information and capture more task-relevant information from unique parts, we can perform conditional masking, where masking is conditioned on augmented views (similar to the multimodal augmentation in the paper, where the conditioned views are approximating the labels). As a result, only unique regions in the second modality can be masked out, making the model capture more unique information from the second modality by masked prediction. Here we have only provided high-level intuitions of extensions to these methods, and future work should explore these ideas in more detail.

# D  Experimental Details

## D.1  Implementation details

**Objective Formulation and Architecture**

In Algorithm 2 in the main text, we see the sketch for doing contrastive learning with our proposed objectives. To implement all algorithms used in our ablation experiments, we start with two encoders $e_1(\cdot)$ and $e_2(\cdot)$, which takes samples $x_1$ and $x_2$ from the modalities $X_1$ and $X_2$, and outputs corresponding representations $z_1$ and $z_2$. We also have a critic function $f_\theta(\cdot, \cdot)$ parametrized by $\theta$ which takes $z_1$ and $z_2$ as inputs and returns a scalar. A popular way to perform contrastive learning aims to maximize $I_{\text{NCE}}(X_1; X_2)$, where

$$I_{\text{NCE}}(X_1; X_2) = \mathbb{E}_{\substack{x_1, x_2^+ \sim p(x_1, x_2) \\ x_2^- \sim p(x_2)}} \left[ \log \frac{\exp f_\theta(e_1(x_1), e_2(x_2^+))}{\sum_k \exp f_\theta(e_1(x_1), e_2(x_2^-))} \right]. \tag{93}$$

In our algorithms, we follow the derivations in Eqs.(8-9) to maximize each $I_{\text{NCE}}$ objective and minimize each $I_{\text{NCE-CLUB}}$ objective. Therefore, for each objective, we add two additional MLP heads on top of the two encoders and create a separate critic which takes in the outputs of the MLP heads instead of the encoders. In all the experiments, we adopt the concat critic design [58, 60, 68], where $f_\theta(x, y) = h_\theta([x, y])$ with $h_\theta$ being an MLP.

**FACTORCL-SUP**: In the supervised version of CL which uses label $Y$, the objective we aim to maximize is formulated as

$$\mathcal{L}_{\text{FACTORCL}-SUP} = I_{\text{NCE}}(X_1; X_2) - I_{\text{NCE-CLUB}}(X_1; X_2|Y) \tag{94}$$
$$+ I_{\text{NCE}}(X_1; Y) + I_{\text{NCE}}(X_2; Y) \tag{95}$$
$$- I_{\text{NCE-CLUB}}(X_1; X_2) + I_{\text{NCE}}(X_1; X_2|Y). \tag{96}$$

Each $I_{\text{NCE}}$ and $I_{\text{NCE-CLUB}}$ term in this objective is calculated using its own critic as discussed above. The conditional terms involving the label $Y$ are implicitly modeled by directly concatenating $Y$ to the outputs of both heads before feeding into the critic. To obtain the learned representations $Z_{S_1}$, we concatenate the outputs of the heads on top of the encoder $e_1$ that correspond to the terms $I_{\text{NCE}}(X_1; X_2)$ and $I_{\text{NCE-CLUB}}(X_1; X_2|Y)$. To obtain $Z_{U_1}$, we concatenate $e_1$'s head outputs from the terms $I_{\text{NCE}}(X_1; Y)$, $I_{\text{NCE-CLUB}}(X_1; X_2)$, and $I_{\text{NCE}}(X_1; X_2|Y)$. $Z_{S_2}$ and $Z_{U_2}$ are obtained in a similar fashion, except we use the outputs from $e_2$'s heads instead of $e_1$.

**FACTORCL-SSL**: In the self-supervised version of CL which uses augmentations $X_1'$ and $X_2'$ of the input modalities, the objective we aim to maximize is formulated as

$$\mathcal{L}_{\text{FACTORCL}-SSL} = I_{\text{NCE}}(X_1; X_2) - I_{\text{NCE-CLUB}}(X_1; X_2|X_1', X_2') \tag{97}$$
$$+ I_{\text{NCE}}(X_1; X_1') + I_{\text{NCE}}(X_2; X_2') \tag{98}$$
$$- I_{\text{NCE-CLUB}}(X_1; X_2) + I_{\text{NCE}}(X_1; X_2|X_1', X_2'). \tag{99}$$

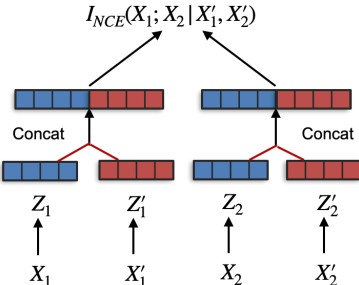

Figure 5: An illustration of conditioning by concatenation in the implementation of FACTORCL. Conditioning is done by concatenating $Z_1$, the encoded representation of $X_1$, and $Z_1'$, the encoded representation of $X_1'$. A similar operation is performed for $X_2$ and $X_2'$. The concatenated vectors are then fed to MI estimators, such as $I_{\text{NCE}}$ and $I_{\text{NCE-CLUB}}$ (the figure illustrates $I_{\text{NCE}}$).

Here the conditional terms are conditioned on the augmentations $X_1'$ and $X_2'$, and we can similarly model it by concatenating the head outputs of $X_1'$ to $X_1$ and the head outputs of $X_2'$ to $X_2$ before feeding into the critic. We use Figure 5 to illustrate this. The way to obtain the learned representations is the same as described in FACTORCL-SUP.

**Estimation of CMI**: To estimate the conditional mutual information (CMI) $I(X_1; X_2|X_1', X_2')$, we can estimate the lower or upper bounds of true CMI [51, 55, 69]. However, direct sampling from the conditional distribution $p(x_1, x_2|x_1', x_2')$ can be expensive because we should consider a different conditional distribution $p(x_1, x_2|x_1', x_2')$ for each data pair $x_1', x_2'$. Sordoni et al. [69] address this by concatenating the conditioning variable with the input in the critic: $\phi(x_1, x_2, c)$, and showing that Conditional InfoNCE (Eq.(15)) is a lower bound and estimator of CMI. This estimator can be made more exact by further importance sampling [69]. However, adding importance sampling [69] or using more accurate estimators [26] comes with a trade-off in complexity. Since we focus on capturing unique information to learn a scalable multimodal representation instead of accurately estimating the CMI, we leveraged a simpler version of the estimator from Sordoni et al. [69]: generating multiple augmented pairs from $x_1, x_2$, and concatenating the input $x_1, x_2$ and each augmented pair $x_1', x_2'$ to define samples from the conditional distribution $p(x_1, x_2|x_1', x_2')$. We argue that since augmentations do not significantly change the semantics of images, $p(x_1, x_2|x_1', x_2')$ could be approximated by $p(x_1'', x_2''|x_1', x_2')$ where $x_1'', x_2''$ are other augmented pairs in addition to $x_1', x_2'$. In this submission, we use one pair of augmented samples for consistency, but our code easily supports increasing the number of augmented pairs that can improve the accuracy of CMI estimation.

Regardless, our existing one-pair implementation can already show that our estimators are empirically comparable to CMI estimators with guarantees such as Mukherjee et al. [55] (Table 9), and our estimators empirically satisfy that the lower bound is smaller than the true CMI, and the true CMI smaller than the upper bound, i.e., Conditional InfoNCE $\leq$ true CMI $\leq$ Conditional InfoNCE-CLUB (also in Table 9). We refer the reader to Sordoni et al. [69] for tighter bounds for CMI.

**OurCL-SUP**: For this ablation, we remove the factorization and only learn $Z_1$ for $X_1$ and $Z_2$ for $X_2$. The objective we use is the same as that of FACTORCL-SUP. The only difference is that we now take $e_1(x_1)$ and $e_2(x_2)$ as the learned representations for inputs $x_1$ and $x_2$.

**OurCL-SSL**: This is a similar ablation for FACTORCL-SSL where we remove the factorization. The objective is the same as that of FACTORCL-SSL and we use $e_1(x_1)$ and $e_2(x_2)$ as the learned representations for inputs $x_1$ and $x_2$.

**Training Strategy**: In regular contrastive learning using $I_{\text{NCE}}$ as the only objective, we can simply perform gradient descent to minimize $I_{\text{NCE}}$, updating all parameters in the encoders, MLP heads, and critics. However, training any of the four methods above also involves the minimization of the $I_{\text{NCE-CLUB}}$ objectives, which require the optimal critic $f^*$ from $I_{\text{NCE}}$, as stated in Eq.(11). Therefore, within each iteration during our training, we need to first obtain the optimal critics for the $I_{\text{NCE-CLUB}}$ terms using the $I_{\text{NCE}}$ objective. We outline the training strategy using a sampling method in Algorithm 3. In this algorithm, $\mathcal{L}_{\text{FACTORCL}}$ can be either $\mathcal{L}_{\text{FACTORCL}-SUP}$ or $\mathcal{L}_{\text{FACTORCL}-SSL}$, and $\mathcal{L}_{\text{NCE}}$ is the summation of $I_{\text{NCE}}$ objectives for the $I_{\text{NCE-CLUB}}$ terms. In particular, we have

$$\mathcal{L}_{\text{NCE}} = \begin{cases} I_{\text{NCE}}(X_1; X_2|Y) + I_{\text{NCE}}(X_1; X_2), & \text{if } \mathcal{L} = \mathcal{L}_{\text{FACTORCL}-SUP} ; \\ I_{\text{NCE}}(X_1; X_2|X_1', X_2') + I_{\text{NCE}}(X_1; X_2), & \text{if } \mathcal{L} = \mathcal{L}_{\text{FACTORCL}-SSL} . \end{cases} \quad (100)$$

---
**Algorithm 3** CL training with sampling
---
**Require:** Multimodal dataset $\{\mathbf{X_1}, \mathbf{X_2}\}$.
---

$\quad \theta, \phi \leftarrow$ Initialize network parameters.
$\quad$ **while** not converged **do**
$\quad\quad \{\boldsymbol{x}_1, \boldsymbol{x}_2\} \leftarrow$ Sampled batch from $\{\mathbf{X_1}, \mathbf{X_2}\}$
$\quad\quad \theta \leftarrow$ Update parameters by maximizing $\mathcal{L}_{\text{FACTORCL}}$
$\quad\quad$ **for** $i = 1$ to $k$ **do**
$\quad\quad\quad \{\boldsymbol{x}_1', \boldsymbol{x}_2'\} \leftarrow$ Sampled batch from $\{\mathbf{X_1}, \mathbf{X_2}\}$
$\quad\quad\quad \phi \leftarrow$ Update parameters by maximizing $\mathcal{L}_{\text{NCE}}$
$\quad\quad$ **end for**
$\quad$ **end while**
$\quad$ **return** $\theta, \phi$

---

We define $\phi$ to be the parameters of critics for the $I_{\text{NCE-CLUB}}$ terms, and $\theta$ corresponds to all the rest parameters in the network (parameters of encoders, heads, and critics for $I_{\text{NCE}}$ terms). In the outer loop, we update $\theta$ using the main learning objective $\mathcal{L}$. In the inner loop, we update $\phi$ using the $\mathcal{L}_{\text{NCE}}$ objective, which learns the optimal critics $f^*$ needed to compute the $I_{\text{NCE-CLUB}}$ terms. Ideally in the inner loop we would update $\phi$ until convergence so we get a good approximation to the optimal critic. In practice we found sampling just one batch by setting $k = 1$ in Algorithm 3 works pretty well. Using only one iteration does not have a big impact on the convergence and still produces promising results. More importantly, it significantly reduces the time required for training, and allows our algorithms to have comparable running time to existing contrastive learning methods.

## D.2  Datasets

**Gaussian datasets for MI estimation**: As shown in Figure 3 in the main text, we first demonstrate the quality of our proposed upper bounds $I_{\text{NCE-CLUB}}(X_1; X_2)$ on a toy Gaussian dataset. We obtain the samples $\{(x_i, y_i)\}$ from 4 multivariate Gaussian distribution with dimensions $\{20, 50, 100, 200\}$. In each dataset, we set the ground truth MI values to be $\{2, 4, 6, 8, 10\}$, and so we can compute the correlation $\rho$ needed for achieving these MI values using the ground truth MI formula for Multivariate Gaussian: $I(X, Y) = -\frac{d}{2} \log(1 - \rho^2)$. At each true MI value we sample 4000 times using a batch size of 64.

**Synthetic dataset with controlled generation**: We generate data with controllable ratios of task-relevant shared and unique information to analyze the behavior of each contrasive learning objective in Figure 1 in the main text. Starting with a set of latent vectors $w_1, w_2, w_s \sim \mathcal{N}(0_d, \Sigma_d^2), d = 50$ representing information unique to $X_1, X_2$ and common to both respectively, the concatenated vector $[w_1, w_s]$ is transformed into high-dimensional $x_1$ using a fixed full-rank transformation $T_1$ and likewise $[w_2, w_s]$ to $x_2$ via $T_2$. The label $y$ is generated as a function (with nonlinearity and noise) of varying ratios of $w_s$, $w_1$, and $w_2$ to represent shared and unique task-relevant information. For experiments, we used 1-layer MLPs with 512 hidden size as encoders, and the embedding dimensions are 128 for both modalities. The heads on top of encoders are also 1-layer MLPs with the same hidden and output dimension as the input, and all critics are 1-layer MLPs with 512 hidden size.

**Multimodal fusion datasets**: We use a collection of 5 real-world datasets provided in Multi-Bench [45] and the IRFL dataset to test our method in the context of varying ratios of shared and unique information that is important for the task. In all the datasets below, the heads added on top of the encoders are 1-Layer MLPs with ReLU activations that map the encoder outputs to the same dimensions. All critics are also MLPs with 1 hidden layer of size 512 and ReLU activation.

1. **MIMIC-III** [38] (Medical Information Mart for Intensive Care III) is a large-scale dataset for healthcare which contains records of over 40,000 ICU patients in both forms of times-series data measured by hours and static data (age, gender, ethnicity) in the tabular numerical form. We use the preprocessed data provided in MultiBench [45], where the time-series data is measured every 1 hour in a 24-hour period and consists of vectors of size 12, and the tabular data consists of vectors of size 5. The task we use in the experiment is a binary classification on whether the patient fits any ICD-9 code in group 7 (460-519).

Table 6: Results on MultiBench [45] datasets with varying shared and unique information: FACTORCL achieves strong results vs self-supervised (top 5 rows) and supervised (bottom 3 rows) baselines that do not have unique representations, factorization, upper-bounds to remove irrelevant information, and multimodal augmentations.

| Model | MIMIC | MOSEI | MOSI | UR-FUNNY | MUSTARD |
|---|---|---|---|---|---|
| SimCLR [13] | 66.7 ± 0.0% | 71.9 ± 0.3% | 47.8 ± 1.8% | 50.1 ± 1.9 % | 53.5 ± 2.9% |
| Cross+Self [81] | 65.2 ± 0.0% | 71.1 ± 0.2% | 48.6 ± 1.2% | 56.5 ± 0.7% | 53.9 ± 4.5% |
| Cross+Self+Fact [89] | 65.5 ± 0.0% | 71.9 ± 0.2% | 49.0 ± 1.1% | 59.9 ± 0.9% | 53.9 ± 4.0% |
| OurCL-SSL | 65.2 ± 0.0% | 71.2 ± 0.2% | 49.0 ± 0.8% | 58.8 ± 1.3% | 54.0 ± 2.5% |
| FACTORCL-SSL | **67.3 ± 0.0%** | **74.5 ± 0.1%** | **51.2 ± 1.6%** | **60.5 ± 0.8%** | **55.8 ± 0.9%** |
| SupCon [41] | 67.4 ± 0.0% | 71.0 ± 0.1% | 47.2 ± 1.2% | 50.1 ± 2.0% | 52.7 ± 2.2% |
| OurCL-SUP | 68.2 ± 0.0% | 71.1 ± 0.2% | 65.3 ± 0.8% | 58.3 ± 1.1% | 65.1 ± 1.8% |
| FACTORCL-SUP | **76.8 ± 0.0%** | **77.8 ± 0.3%** | **69.1 ± 0.6%** | **63.5 ± 0.8%** | **69.9 ± 1.9%** |

Table 7: Continued pre-training on CLIP with our FACTORCL objectives on classifying images and figurative language. Our approach shows strong results as compared to standard fine-tuning and continued pre-training.

| Task | IRFL |
|---|---|
| Zero-shot CLIP [61] | 89.2 ± 0.0% |
| SimCLR [13] | 91.6 ± 0.0% |
| Cross+Self [81, 89] | 91.1 ± 1.2% |
| FACTORCL-IndAug | 91.6 ± 1.3% |
| FACTORCL-SSL | **93.8 ± 1.4%** |
| Fine-tuned CLIP [61] | 96.4 ± 0.0% |
| SupCon [41] | 87.7 ± 4.7% |
| FACTORCL-SUP | **98.3 ± 1.2%** |

In the experiments, we use a 2-layer MLP with 10 hidden layer size for the tabular data modality, and map it to a vector of size 10. The time-series modality is encoded using a GRU with hidden size 30 and followed by a linear layer which projects the output to embeddings of size 15. We train the model for 100 epochs using the Adam optimizer with a 1e-4 learning rate.

2. **CMU-MOSEI** [93] is the largest sentence-level multimodal sentiment and emotion benchmark with $23,000$ monologue videos. It contains more than 65 hours of annotated video from more than 1,000 speakers and 250 topics. Each video is labeled with a sentiment intensity ranging from -3 to 3. In our experiments, we cast the intensity values to a binary classification on whether the sentiment is positive or negative. MultiBench [45] provides access to the extracted features of the vision, text, and audio modalities, and in our experiments, we use the vision and text features for doing contrastive learning.

   In our experiments, we encode both the vision and text modalities using Transformer encoders with 5 heads and 5 layers, and map them to 40-dimensional embeddings. We train the model for 100 epochs using the Adam optimizer with a 1e-4 learning rate.

3. **CMU-MOSI** [91] is a similar dataset for multimodal sentiment analysis created from $2,199$ YouTube videos clips. The data focuses on videos that reflect the real-world distribution of speakers expressing their opinions in the form of monologues. The sentiment intensities are labeled continuously from -3 to 3. Again we cast the label into a binary classification on whether the sentiment is positive or negative, and we used the extracted vision and text features for contrastive learning.

   In our experiments we encode both the vision and text modalities using Transformer encoders with 5 heads and 5 layers, and map them to 40-dimensional embeddings. We train the model for 100 epochs using the Adam optimizer with a 1e-4 learning rate.

4. **UR-FUNNY** [27] is the first large-scale dataset for humor detection in human speech. The dataset consists of samples from more than $16,000$ TED talk videos with speakers from diverse backgrounds sharing their ideas. The laughter markup is used to filter out 8,257 humorous punchlines from the transcripts. The context is extracted from the prior sentences to the punchline. Using a similar approach, 8,257 negative samples are chosen at random intervals where the last sentence is not immediately followed by a laughter marker. The task is to classify whether there is humor or not using the vision and text modalities.

Table 8: Additional experiments on CIFAR10 [43] and MNIST [19] datasets using our FACTORCL objectives on image classification.

| Task | CIFAR10 | MNIST |
|---|---|---|
| SimCLR [13] | 87.0% | 98.84% |
| SupCon [41] | 92.7% | 99.38% |
| FACTORCL-SUP | 91.3% | 99.21% |

Table 9: We verify our conditional lower and upper bound estimators on a synthetic dataset with fixed dimension of representation $d_z$ and varying number of samples $n$.

| Number of samples ($\times 10^3$), $d_z = 20$ | 5 | 10 | 20 | 50 |
|---|---|---|---|---|
| CCMI (MI-Diff + Classifier) | 2.03 | 2.06 | 2.15 | 2.20 |
| Conditional InfoNCE | 2.19 | 2.20 | 2.20 | 2.20 |
| Conditional InfoNCE-CLUB | 3.45 | 3.53 | 2.98 | 2.86 |
| True CMI | 2.32 | 2.32 | 2.32 | 2.32 |

In our experiments, we encode both the vision and text modalities using Transformer encoders with 5 heads and 5 layers, and map them to 40-dimensional embeddings. We train the model for 100 epochs using the Adam optimizer with a 1e-4 learning rate.

5. **MUSTARD** [12] is a corpus of 690 videos for research in sarcasm detection from popular TV shows including Friends, The Golden Girls, The Big Bang Theory, and Sarcasmaholics Anonymous. It contains audiovisual utterances together with the textual context. We use the preprocessed features of the vision and text modalities for doing contrastive learning and performing sarcasm detection.

In our experiments, we encode both the vision and text modalities using Transformer encoders with 5 heads and 5 layers, and map them to 40-dimensional embeddings. We train the model for 100 epochs using the Adam optimizer with a 1e-4 learning rate.

6. **IRFL** [88] is a dataset for understanding multimodal figurative languages. It contains $6,697$ matching images and figurative captions (rather than literal captions) of three types of figurative languages: idiom, simile, and metaphor. The original data for the matching task is provided in the form of 1 caption, 3 distractor images, and 1 matching image. We convert it into a fusion task by only collecting the matching image and text pairs and assigning labels using the type of figurative language it belongs to.

For this dataset, we do not train from scratch. Instead, we performed continued pretraining using our proposed objectives on pretrained CLIP [61] models. We used the CLIP-VIT-B/32 model and its pretrained image and text encoders. We performed training for 10 epochs using the Adam optimizer with a 1e-6 learning rate.

### D.3 Additional analysis and results

**Fusion experiments**: In Table 6 and 7 we present more detailed results on the Multibench [45] and IRFL [88] datasets computed from 5 independent runs. FACTORCL significantly outperforms the baselines that do not capture both shared and unique information in both supervised and self-supervised settings, particularly on MUSTARD (where unique information expresses sarcasm, such as sardonic facial expressions or ironic tone of voice), and on MIMIC (with unique health indicators and sensor readings). There are also big improvements on the two sentiment analysis datasets MOSEI and MOSI, with $6.8\%$ and $21.9\%$ increases respectively when compared to SupCon [41].

In Table 7, we also see that FACTORCL substantially improves the state-of-the-art in classifying images and figurative captions which are not literally descriptive of the image on IRFL, outperforming zero-shot and fine-tuned CLIP [61] as well as continued pre-training baselines on top of CLIP. While the supervised version gives the best results overall, the self-supervised version with our proposed unique augmentations also performs better than independent augmentations, indicating that in the case without label information, we should always try to find and use unique augmentations when possible. In our experiments, we use word masking for text augmentations. For independent image

Table 10: We verify our conditional lower and upper bound estimators on a synthetic dataset with varying dimensions of representation $d_z$ and fixed number of samples $n$.

| Dimension $d_z$, $n = 2 \times 10^4$ | 1 | 10 | 20 | 50 | 100 |
|---|---|---|---|---|---|
| CCMI (MI-Diff + Classifier) | 2.30 | 2.18 | 2.15 | 1.98 | 1.67 |
| Conditional InfoNCE | 2.18 | 2.20 | 2.20 | 2.26 | 2.30 |
| Conditional InfoNCE-CLUB | 3.70 | 2.95 | 2.98 | 2.79 | 2.86 |
| True CMI | 2.32 | 2.32 | 2.32 | 2.32 | 2.32 |

Table 11: We probe whether the InfoMin assumption from Tian et al., $I(Z_1; Y|X_2) = 0$ and $I(Z_2; Y|X_1) = 0$, is reasonable for Theorem 1. Compared to the shared information $I(X_1; X_2)$, $I(Z_1; Y|X_2)$ is much smaller and closer to zero, indicating that the InfoMin assumption is reasonable, and Theorem 1 holds in practice.

| $I(X_1, Y; X_2)$ | $I(X_1; X_2)$ | $I(Z_1; Y|X_2)$ | $I(X_2, Y; X_1)$ | $I(X_2; X_1)$ | $I(Z_2; Y|X_1)$ |
|---|---|---|---|---|---|
| 12.69 | 12.29 | 0.40 | 11.31 | 10.92 | 0.38 |

augmentations, we use cropping, flipping, and color jittering. The unique augmentation simply removes the cropping operation, as illustrated in Figure 4 in the main text.

**Additional experiments on high shared information and low unique information**: In Table 8 we include additional results using our method on the CIFAR10 [43] and MNIST [19] datasets. Our method outperforms the self-supervised contrastive learning on both datasets as expected, and roughly maintains the same performance as supervised contrastive learning. Therefore, in cases with abundant shared information (two modalities with high shared information or two different views generated from augmentations), our method recovers the performance of existing methods that do not capture unique information.

**Experiments on CMI estimator verification**: In Table 9 and Table 10, we include experiment results which verify that computing the conditional MI lower and upper bounds via concatenation indeed yields reliable estimates. In particular, we aim to verify that the the Conditional InfoNCE objective gives a lower bound of the CMI, and the Conditional InfoNCE-CLUB objective gives an upper bound of the CMI. We follow the experiment setups in [55], presenting the true CMI and results of our estimators on synthetic data with fixing dimension of representation $Z$ and varying samples $n$, and fixing samples $n$ and varying $d_z$. The specific implementations used for conditional InfoNCE and conditional InfoNCE-CLUB can be found in Equation 13 and Equation 14, respectively. The results indicate that our Conditional InfoNCE gives estimations smaller than the true CMI, and Conditional InfoNCE-CLUB gives estimations greater than the true CMI. The performances are comparable to estimators in [55], suggesting that our method yields valid and competitive lower and upper bounds for CMI.

**Empirical verification on InfoMin assumption**: To verify the InfoMin assumption [72] ($I(Z_1; Y|X_2) = I(Z_2; Y|X_1) = 0$), we use the same synthetic dataset as in Table 1 and measure $I(Z_1; Y|X_2)$. The results are shown in Table 11: we get $I(X_1; X_2) = 12.29$ and $I(Z_1; Y|X_2) = 0.4$. $I(Z_1; Y|X_2)$ is much smaller and closer to zero than $I(Z_1; Y|X_2)$, indicating that the InfoMin assumption holds in practice.

**Compute resources**: All experiments in this paper are run on a single NVIDIA A100 GPU. It takes about 10 to 12 GPU hours to train the model on the CIFAR10 [43] for 300 epochs, and all the other experiments can be finished within 1 GPU hour using our specified hyperparameters.

