# OpenReview forum: "Factorized Contrastive Learning: Going Beyond Multi-view Redundancy"
_NeurIPS.cc/2023/Conference — NeurIPS 2023 poster_

### Official Review · Reviewer_N8Dx · 2023-06-27

**Soundness:** 3 good
**Presentation:** 3 good
**Contribution:** 3 good
**Rating:** 6
**Confidence:** 3

**Summary:**

The paper introduces FactorCL, a novel method for learning multimodal representations. The proposed method generalizes traditional MI maximization-based approaches by capturing both shared and unique information relevant to downstream tasks. Based on the information-theoretic perspective, the paper demonstrates that traditional CL approaches, based on shared task-relevant information maximization, perform sub-optimally in the presence of non-shared task-relevant information, which is an easily imaginable case in a multi-modal setting. From here, the paper proposes a novel CL objective and augmentation strategy to address the uniqueness gap. Experimental evaluation demonstrates that the proposed method outperforms several existing CL approaches in multi-modal setting.

**Strengths:**

1. The work is very timely. The paper addresses an important question of adopting CL methods for multi-modal scenarios.
2. Paper is clearly written and easy to follow.
3. The proposed method can be seen as an information-theoretic generalization of base contrastive learning approaches, which only address shared task-relevant information. This is a natural extension.
4. Section 2 provides an intuitive information-theoretic analysis of multi-modal CL. Figure 2 gives a nice overview of the bounds and their role in the objective.
5. Experimental results clearly demonstrate the benefits of FactorCL.

**Weaknesses:**

1. If the information is modality "unique", but task-relevant, shouldn't it end up in the shared task-relevant partition by means of simply optimizing for the shared information (as non-shared info is implicitly minimized this way) as standard CL methods do?
2. If weakness 1 holds, then what is the reason why standard CL methods fail? Can it be solely attributed to sub-optimal augmentation, which limits the shared information?
3. Table 2 reveals that most of the performance increase can be attributed to improved augmentation. Is it connected to Weaknesses 1 & 2?
4. L 31-39. Isn't low shared information a direct consequence of high unique information (and vice versa)? If yes, then there is no need to define it as a separate imitation.

Minor:
 - Figure 2 typo X1 (modality 2)
 - Eq (1) typo I(x1, x2, y)  -> I(x1, x2 | y)
 - Formatting of Table 2 can be improved, the top row X1;X2 is hard to read
 - L171-174 is confusing. Does it mean X1 and X2 are concatenated with the encoded features of (X1' & X2')?

**Questions:**

My questions are in Weaknesses 1,2,3,4.

**Limitations:**

I suggest the paper include a separate section on the limitations of the methods, where authors elaborate on challenging scenarios, where they expect the proposed approach to not deliver substantial improvements.

---

> ### Author Rebuttal · Authors · 2023-08-10
>
> Thank you for your valuable feedback and insightful comments! We respond to some concerns below:
>
> [Unique partition] We define shared information as $I(X_1; X_2; Y)$ and unique information as $I(X_1; Y | X_2)$ and $I(X_2; Y | X_1)$ based on information theory, and from this definition the two areas of shared and unique information are strictly disjoint. **Note that multimodal and multiview are different: in multimodal, the $X_1$ and $X_2$ are fixed, so the shared and unique regions are fixed. In multiview, a single input $X$ is augmented into $X$’, so there are ways to control how much shared and unique information there is between $X$ and $X’$ [Tian et al., InfoMin].**
>
> In practice, Theorem 1 shows the result of CL on multimodal data - it will discard unique information according to our definition. Figure 1 shows this in practice, with a gradual decrease of standard CL in performance as the ratio of unique information increases.
>
> [Is Weakness 1 true, and why does standard CL fail] Weakness 1 does not hold, as we explained in the answer above. Why standard CL fails is not solely due to sub-optimal augmentation but is a fundamental limitation of standard CL methods - **contrasting between $(X_1, X_2)$ pairs can only capture shared mutual information between $X_1$ and $X_2$, but not unique information regardless of how augmentation is performed. Unique information has to be captured via modality-specific signals, alongside the right modality-specific augmentations and factorized representations, which we show in FactorCL.**
>
> [Performance improvement] The Table 2 performance increase is not only from better augmentation. Both FactorCL-SUP and Factor-SSL have increased performance, but improved augmentation is only used in the FactorCL-SSL. Performance increases from Table 2 come from our method being able to unique task-relevant information in both supervised and SSL, directly addressing limitation 2 in the introduction.
>
> [Low shared and high unique] High unique information may not imply low shared information. There could be cases where both shared information and unique information are high. The MOSEI dataset [83] is an example where modality-unique information (text expressing a joyful event) and modality-shared information (happiness from a delighted voice and a smiling face) are both rich and useful for downstream sentiment/emotion tasks.
>
> [Minor comments] We thank the reviewers for the suggestions. We fixed the Figure 2 typo. The Eq. 1 does not have a typo, and $I(X_1; X_2; Y)$ is the correct form of the shared information. We have improved the formatting of Table 2. And yes, Lines 171-174 mean $X_1$ and $X_2$ are concatenated with the encoded features of ($X_1'$ & $X_2'$)
>
> [Limitations] We will add more discussions regarding limitations to Appendix:
> 1. Optimizing better MI lower and upper bounds could improve performance for higher-dimensional and complex modalities.
> 2. Extending the work of InfoMin [Tian et al.] to automatically generate data augmentations to satisfy the optimal augmentation assumption, and leveraging future progress in multimodal generative models for data augmentation.
> 3. Quantifying whether shared or unique information is more important for the task to optimize FactorCL better.

---

> > ### Comment · Reviewer_N8Dx · 2023-08-16
> >
> > I thank authors for the response. The rebuttal addresses my concerns, given the clarifications are added to the main paper. After reading the rebuttal and other reviews, I keep my score at weak accept.

---

### Official Review · Reviewer_4hLy · 2023-07-05

**Soundness:** 3 good
**Presentation:** 3 good
**Contribution:** 3 good
**Rating:** 4
**Confidence:** 4

**Summary:**

This work addressed the problem of contrastive learning in a multimodal setting, particularly in capturing shared and modality-specific information regarding downstream tasks. Existing approaches assume that the information contained in different modalities is somewhat the same (redundant), but in the real world, it might not be the case. The authors proposed to factor the task-specific mutual information into shared and unique components. They provided a derivation for this factorization and their FactorCL algorithm optimized these two parts with approximated upper and lower bounds separately, under an assumption of optimal unimodal augmentation. The authors evaluated against chosen baselines on a collection of tasks in the MultiBnech benchmark and show improvements over these baselines.

**Strengths:**

- The authors provided a lot of details about their approaches, from the analysis of CL algorithms, and their approximation to experiment results.
- Most of the simplifications look intuitively reasonable
- On the chosen benchmark, the gains by their algorithm were decent


**Weaknesses:**

- The paper is packed with too many details which makes it hard to follow the main idea
- On the experiment results, for example in Table 2, even though there were improvements over baselines, the results were too far below those reported in previous works, including the original MultiBench paper (https://arxiv.org/pdf/2107.07502.pdf - also Table 2). This casts doubt on the validity of the approach of this paper.

**Questions:**

- why don't you conduct experiments on various well-established vision language image-text datasets out there such as MS-COCO or Flickr30K?

---

> ### Author Rebuttal · Authors · 2023-08-10
>
> Thank you for your valuable feedback and insightful comments! We respond to some concerns below:
>
> [Too many details] The details cover the exact mathematical derivation from the definitions of shared and unique task-relevant information to our final self-supervised objectives. We will add an overview of the derivation steps and intermediate objectives and move some details to Appendix for clarity. **We also emphasize that the actual implementation is relatively straightforward to adapt to a standalone contrastive framework (Algorithms 2 and 3). Code is included in the supplementary, and we will release it publicly so that all implementation details are fully transparent and reproducible.**
>
> [Performance gap] There are 2 key differences between the experiments:
> 1. Our approach is currently implemented for two modalities as we primarily study two views in our framework, while the best reported MultiBench results are for supervised learning with three modalities.
> 2. MultiBench and related work design the best modality and task-specific multimodal architectures to really push for the best supervised learning performance (e.g., complex multimodal transformer methods), while we use generic encoders (not necessarily the most complex architectures) and focus on contrastive representation learning objectives, and use linear probing to evaluate performance. This is in line with observations where SSL with generic encoders (e.g., SimCLR) can have lower accuracies than supervised models whose architecture is optimized for supervised performance (e.g., CoAtNet [Dai et al.]) and due to linear probing evaluation (e.g., Chen et al.).
>
>
> During the rebuttal period, we scaled up some experiments to address the differences due to points 1 and 2 and summarize the results in Table 2 in the rebuttal pdf:
> 1. We use the architecture from the best supervised models, and we first pretrain the model using FactorCL-SSL and then perform supervised fine-tuning to evaluate. The total number of epochs for FactorCL-SSL pretraining equals the total number of epochs for training supervised baselines, with FactorCL-SSL having a few epochs (less than ten) for fine-tuning. We also extended FactorCL to three modalities: we first perform vision + language FactorCL-SSL pretraining and concatenate audio features for supervised fine-tuning. The number of fine-tuning epochs is also less than ten.
> 2. **Overall, we see better results from Factor-CL (ours) than the supervised baselines we reproduced across different datasets as well as architectures**, suggesting that our method achieves stronger results by capturing unique task-relevant information. There is still a gap between the FactorCL-SSL results and the reported supervised results in Multibench; we attribute this small gap to the fact that the supervised models in Multibench use all three modalities to train from scratch, while we only use the audio modality to fine-tune with a few epochs, so it does not learn all multimodal interactions.
>
> Dai et al. Coatnet: Marrying convolution and attention for all data sizes. NeurIPS 2021.
>
> Chen et al. Big self-supervised models are strong semi-supervised learners. NeurIPS 2020.
>
> [Datasets] These established vision language retrieval benchmarks test only for shared information between images and captions and do not need unique information.
>
> **During the rebuttal period, we further added experiments on the NYCaps dataset** [Hessel et al.], a new dataset testing the matching of cartoon images and humorous captions, which requires unique humorous information in images and text. We show strong performance when continuing contrastive training on top of a pre-trained CLIP using SimCLR, SupCon, or FactorCL and evaluating using zero-shot retrieval, **SimCLR has an accuracy of 49.2%, SupCon has an accuracy of 49.7%, and our method outperforms both by achieving 50.5%.**
>
> Our paper also includes experiments on IRFL [78], which aims to match images and figurative captions (rather than literal captions), requiring more unique information. From Table 3, we outperform the state-of-the-art in classifying images and figurative captions, outperforming zero-shot, fine-tuned, and continued pre-trained CLIP models.
>
> Hessel et al., Do Androids Laugh at Electric Sheep? Humor "Understanding" Benchmarks from The New Yorker Caption Contest. ACL 2023.

---

### Official Review · Reviewer_bfCW · 2023-07-05

**Soundness:** 3 good
**Presentation:** 4 excellent
**Contribution:** 3 good
**Rating:** 7
**Confidence:** 4

**Summary:**

Based on the mutual-information theory, this paper proposes a new multi-modal contrastive learning method (FactorCL) to learn both shared and unique multi-modal task-relevant information, which captures task-relevant information via maximizing MI lower bounds and removing task-irrelevant information via minimizing MI upper bound. The method achieves significant influence in different real-world datasets.

**Strengths:**

1. How to capture both the shared and unique task-relevant information in different modalities is interesting.
2. The theoretical analysis looks solid and cooperates well with empirical results.
3. The writing is well and easy to follow. Especially the figures represent the main ideas and are easy to understand.
4. The empirical results show significant improvements on different types of datasets.

**Weaknesses:**

1. It seems that the unique-augmentation step is strongly related to the task-relevant information of different modalities. For example, when the text is “a yellow flower”, the ColorJitter operation should be removed. And when the text is “a flower”, the ColorJitter operation is a useful data augmentation. Is it possible to design a strategy that can automatically select the appropriate data augmentations?
2. As shown in Table 2, FactorCL-SSL and FactorCL-SUP shows significant difference on some datasets. Especially, on MOSI, SimCLR and Supcon show similar performance while the performance of FactorCL-SSL and FactorCL-SUP have a large gap. Is it possible to provide more discussions about that?

**Questions:**

See weaknesses.

---

> ### Author Rebuttal · Authors · 2023-08-10
>
> Thank you for your valuable feedback and insightful comments! We respond to some concerns below:
>
> [Automatic augmentations] We find that **augmentations that approximately satisfy the optimal multimodal augmentation defined in Eqs.17-18 are sufficient for good performance, which is simpler and straightforward to implement on real-world datasets** than strictly satisfying Eqs.17-18. Our intuition is to avoid augmentations that remove or strongly destroy information shared by the other modality (e.g., the caption) and augment via cropping or color jittering in the image. We will clarify this in the paper as a guideline for practitioners. From Table 3, we outperform independent augmentations and other baselines.
>
> It is possible to extend the work of InfoMin [Tian et al.] to generate data augmentations to automatically satisfy the optimal augmentation assumption. Still, these methods require generative models for multiple modalities. We think this is an exciting direction for future work and expect progress in multimodal generative models to yield further advances in these problems.
>
> [Discussion on the gap between FactorCL-SSL and FactorCL-SUP] We found that training a generic multimodal model with SSL was difficult due to the small sample size and high-dimensional and temporal challenges of the MOSI dataset. During rebuttal, we adapted the best supervised multimodal architecture for MOSI. We performed CL pre-training on this architecture, which yielded an improved FactorCL-SSL performance of 80%, closer to FactorCL-SUP performance. We include these updated results and comparisons in Table 2 of the attached rebuttal pdf.

---

> > ### Comment · Reviewer_bfCW · 2023-08-20
> > **reply to the rebuttal**
> >
> > I thank the authors for the responses. I will maintain my original rating.

---

### Official Review · Reviewer_859W · 2023-07-05

**Soundness:** 3 good
**Presentation:** 3 good
**Contribution:** 3 good
**Rating:** 6
**Confidence:** 4

**Summary:**

This paper presents FACTOR CL, a method for multimodal representation learning that captures both shared and unique task-relevant information, going beyond the common approach of focusing on shared information across different data modalities. FACTOR CL is based on three key contributions: factorizing task-relevant information into shared and unique representations, capturing and discarding task-relevant and irrelevant information through optimizing mutual information bounds, and utilizing multimodal data augmentations to approximate task relevance in the absence of labels. On large-scale real-world datasets, FACTOR CL achieves state-of-the-art results on six benchmarks. The premise of this paper is quite similar to the following work (Chaudhuri et al., Cross-Modal Fusion Distillation for
Fine-Grained Sketch-Based Image Retrieval: https://bmvc2022.mpi-inf.mpg.de/0499.pdf), however, the formulation, experimentations are different.


**Strengths:**

(1) This method is innovative in its attempt to model not only the shared information but also the unique information between different modalities, which is often overlooked by traditional Contrastive Learning (CL) methods. The framework is applicable to a broad range of settings, not just those where shared information is dominant. This allows it to handle diverse multimodal distributions, even those with substantial unique information.

(2) The proposed method utilizes self-supervised data augmentations, which allows it to learn task-relevant information without access to labels. This enables the algorithm to learn in a more unsupervised manner, reducing the need for extensive labeled data.

**Weaknesses:**

(1) The method relies on certain assumptions, such as the optimal augmentation assumption, which may not always hold in real-world applications. If these assumptions do not hold, the effectiveness of the method may be significantly compromised. The method's performance might heavily rely on the quality of the data augmentation techniques used, which might not always be easy to decide or optimize.

(2) Calculating mutual information can be challenging in practice, especially when dealing with high-dimensional data. Though approximations are used, they may not always accurately capture the true mutual information.

(3) The method involves complex formalism and numerous mathematical assumptions, which could make it difficult to implement and adapt to different scenarios.

**Questions:**

(1) I wonder what are the limitations of current contrastive learning (CL) methods in modeling unique information in the context of multimodal data.

(2) It would be nice to clarify how the proposed method distinguishes between shared and unique information across different modalities. What does the 'uniqueness gap' in this context mean? How does it affect the learning process?

(3) How does the proposed framework handle situations where there is little to no shared information between modalities, or where most of the shared information is irrelevant to the task?

(4) In the derivation of supervised contrastive learning objectives, how do the lower and upper bounds for mutual information terms impact the learning process?

(5) What are the practical implications of applying semantic augmentations on each modality in the context of unsupervised contrastive learning?

(6) How does the method ensure the learning of task-relevant information without access to labels in the case of unsupervised contrastive learning?


**Limitations:**

The authors have not adequately addressed the limitations of the proposed method. Potential limitations could be found from my comments listed under the weaknesses and limitations sections.

---

> ### Author Rebuttal · Authors · 2023-08-10
>
> Thank you for your valuable feedback and insightful comments! We respond to some concerns below:
>
> [Assumption] Augmentations that exactly satisfy $I(X_1; X_1') = I(X_1; Y)$ and $I(X_2; X_2'|X_1) = I(X_2; Y|X_1)$ (Eqs.17-18) are hard, so instead we relax it to $I(X_1; X_1') \approx I(X_1; Y)$ and $I(X_2; X_2'|X_1) \approx I(X_2; Y|X_1)$, by using these intuitions to guide augmentation design (details in Appendix D.3 and examples in Table 5): our intuition is to avoid augmentations that will remove or strongly destroy information shared by the other modality (e.g., the caption), and augment via cropping or color jittering in the image. We will clarify this in the paper as a guideline for practitioners.
> **Our augmentations consistently perform better than independent augmentations (Table 3), suggesting that approximately satisfying this condition is sufficient**. Most importantly, the unique augmentations are simple and scalable to real-world datasets.
>
> [Estimator accuracy] Our goal is not to calculate MI precisely but to derive lower and upper MI bounds to design objectives that scale on real-world datasets, making training Factor-CL on large datasets possible. Our work provides opportunities for future work to integrate tighter MI estimators (e.g., Guo et al.) to capture task relevance.
>
> Guo et al. Tight mutual information estimation with contrastive fenchel-legendre optimization. NeurIPS 2022.
>
> [Complex math] The details cover the exact mathematical derivation from shared and unique task-relevant information definitions to our final self-supervised objectives. For clarity, we will add an overview of the derivation steps and intermediate objectives. **We also emphasize that the actual implementation is relatively straightforward to adapt to a standalone contrastive framework (Algorithms 2 and 3). Code is included in the supplementary, and we will release it publicly so that all implementation details are fully transparent and reproducible.**
>
> [Limitations of CL] Theorem 1 shows that current CL methods only keep shared information and discard unique information from modalities. This makes representations of standard CL sub-optimal because task-relevant information from uniqueness is not captured. This is also empirically supported by Figure 1, where the downstream performance degrades consistently as unique information increases, suggesting the sub-optimality of standard CL. Table 2 also shows that standard CL’s failure to capture unique information leads to inferior performances on real-world datasets. These are fundamental limitations of CL methods that require new learning paradigms to solve.
>
> [Shared and unique info, uniqueness gap] We define shared information as $I(X_1; X_2; Y)$, unique information as $I(X_1; Y | X_2)$ and $I(X_2; Y | X_1)$, and uniqueness gap as $I(X_1, X_2; Y) - I(Z_1, Z_2; Y)$. The uniqueness gap measures the difference of task-relevant information between input ($X_1, X_2$) and encoded representation ($Z_1, Z_2$). Standard CL will have this gap because current CL methods only keep shared information and discard unique information. In practice, datasets such as IFRL in Table 3 may have high task-relevant unique information, which may widen this gap in standard CL, as discussed in Lines 36-39. In the proposed Factor-CL, the optimal representations will close this gap by capturing both shared and unique info, as shown in Theorem 3.
>
> [No shared or shared is task-irrelevant] Our method handles both of these cases:
> 1. If there is little shared information, our method will still capture the shared information by the lower bound in Eq. 8 and unique information by the lower bounds in Eq 9.
> 2. If most shared information is irrelevant - our method will discard the irrelevant shared information by the upper bound in Eq. 9 and keep the relevant shared information by the lower bound in Eq. 8.
>
> [How bounds impact learning] Our derived lower and upper bounds approximate the actual shared and unique information regions intractable to compute exactly. Our bounds are closer to true MI than existing ones (NCE and CLUB), as shown in Figure 3. Optimizing these tight lower and upper bounds enables us to learn representations to capture different information regions efficiently.
>
> Factorization is also important - each representation optimizes different lower or upper bound objectives, so we can capture the correct information to satisfy multiple terms simultaneously. This makes training easier and performance better (Table 2 and Lines 287-289).
>
> [Practical implications] Besides the implications for multimodal augmentation (Lines 189-196), we discuss augmentation intuitions for each individual modality. For vision, if downstream tasks are object-oriented, the implication is to augment task-irrelevant parts only (e.g., non-object semantic parts if the task is object-centric) or to avoid image augmentations that hugely remove information (e.g., cropping and greyscale). For text, we should try to avoid masking large chunks of text since this can hugely remove information. We want to emphasize that these choices are scalable and much less expensive than labeling data.
>
> [How to ensure learning task-relevant information without labels] We discuss this in section 3.2, specifically definitions 2 and 3. Essentially we design suitable augmentations of each modality, extending the work in [62], and replace Y with X’ (augmented data view) to enable the learning of task-relevant information without access to labels. The details can be found in Lines 157-174.
>
> [Limitations] We will add more discussions regarding limitations:
> 1. Optimizing better MI lower and upper bounds for higher-dimensional and complex modalities.
> 2. Extending the work of InfoMin [Tian et al.] to automatically generate data augmentations to satisfy the optimal augmentation assumption, and leveraging progress in generative models for data augmentation.
> 3. Quantifying shared and unique information to optimize FactorCL better.

---

> > ### Comment · Reviewer_859W · 2023-08-16
> > **Rebuttal response**
> >
> > I thank the authors for the for submitting the rebuttal. However, the optimal augmentation assumption is not getting clear, where Appendix D.3 is not helping much. Furthermore, different parts of the paper are quite complex to understand. So, I am leaning to keep my original rating at the moment.

---

> > > ### Author Response · Authors · 2023-08-18
> > > **Clarification on the optimal augmentation assumption**
> > >
> > > We thank the reviewer for the kind response, we would like to further clarify the optimal augmentation assumption with the example in Figure 4 of the paper:
> > >
> > > – $x_1$ = Image: a car speeding on the highway.
> > >
> > > – $x_2$ = Figurative caption: “The car is as fast as a cheetah.”
> > >
> > > – $y$ = 1, indicating the figurative description and image is a match.
> > >
> > >
> > > ***Task-relevant information:***
> > >
> > > Shared info: car speeding / the car is fast.
> > >
> > > Unique info in $x_1$: highway.
> > >
> > > Unique info in $x_2$: cheetah.
> > >
> > >
> > > ***Independent augmentation (Tian et al.; Eq. 17 in our text):***
> > >
> > > – Task-relevant info in $x_1$: car, speeding, highway.
> > >
> > > To augment the image such that $I(x_1, x_1’) = I(x_1, y)$, we randomly remove image pixels that are not in the car, the speeding lines, or the highway.
> > >
> > > – Task-relevant info in $x_2$: car, fast, cheetah.
> > >
> > > To augment the caption such that $I(x_2, x_2’) = I(x_2, y)$, we randomly mask words except for the words: car, fast, and cheetah.
> > >
> > > ***Optimal unique augmentation (ours, Eq. 17 and 18 in our text):***
> > >
> > > — Task-relevant unique info in $x_1$, $I(x_1,y|x_2)=$ highway.
> > >
> > > To augment the image such that $I(x_1, y|x_2) = I(x_1, x_1’|x_2)$, we only augment image pixels that are not the highway.
> > >
> > > — Task-relevant unique info in $x_2$, $I(x_2, y | x_1)=$ cheetah.
> > >
> > > To augment the caption such that $I(x_2, y | x_1) = I(x_2, x_2’ | x_1)$, we randomly mask words except for the word cheetah.
> > >
> > > While these might not be easy to satisfy exactly, we find empirical solutions that work well to approximate the optimal unique augmentation:
> > >
> > > ***Approximate unique augmentation***
> > >
> > > To approximately augment the image such that $I(x_1, y | x_2) \approx I(x_1, x_1’ | x_2)$, we apply simple rotation augmentations to make the highway pixels remain intact.
> > >
> > > To approximately augment the caption such that $I(x_2, y | x_1) \approx I(x_2, x_2’ | x_1)$, we simply mask or replace non-object words to make the word cheetah remains intact.
> > >
> > > Again the feedback is truly appreciated, and please let us know if you have further questions.

---

### Official Review · Reviewer_fNpT · 2023-07-07

**Soundness:** 2 fair
**Presentation:** 3 good
**Contribution:** 3 good
**Rating:** 6
**Confidence:** 4

**Summary:**

The goal of the proposed work is control over the information content of representations.  The most common form of contrastive learning leverages multi-view redundancy, where data points  are paired across different modalities or different augmentations in the same modality, and a contrastive loss is used to extract the redundant (or shared) information across the pairs.  The authors propose a methodology to extract the shared information and more---the information that is unique to each modality and still relevant to the task---with each component of the information in its own representation.  While contrastive learning based on multi-view redundancy involves maximizing a lower bound on the mutual information (e.g., InfoNCE), the proposed method requires multiple information terms, including the minimization of certain ones, for which the authors propose an efficient scheme of upper and lower bounds.  For scenarios where the task is not given, the task-relevant unique information is defined through a principled augmentation scheme.  The proposed method is benchmarked on a synthetic experiment, where the information content in each variable can be controlled, and then on several multi-modality datasets, where the goal is simply test set performance.

**Strengths:**

The proposed work is well-motivated.  Control over the specific information content of learned representations is a valuable research pursuit, and the information theoretic analysis is reasonable.  The practicality of the methodology is an important strength: standard contrastive learning losses are all one needs, and the MI upper bounds are a clever re-utilization of critics trained concurrently for MI lower bounds.  The strategy of principled data augmentation as a replacement for task information is interesting.  Overall, the ideas pursued in this work have substantive originality and significance.

**Weaknesses:**

While the ideas pursued in this work are good, there are some issues that should be remedied before publication.
- Most importantly, there doesn’t seem to be any support for the way in which the proposed work estimates conditional MI by concatenating the conditional variable(s) to the critic inputs.  This is a significant issue, as both objectives in the information decomposition rely on estimation of the conditional mutual information.  The details are not given (The Appendix refers to a nonexistent section: “In this work, we implement the conditioning in $p(x_1, x_2∣x_1^\prime, x_2^\prime)$ through concatenation and the details are in Appendix Sec.” (Appendix, L720).  Instead there is a seemingly irrelevant discussion of a kernel-based alternative to what is actually done.
- As currently written, Theorem 1 is incorrect, as shown by a simple counterexample.  The identity transform ($Z_1=X_1$, $Z_2=X_2$) “perfectly maximizes” eqn 2, and gives $I(Z_1,Z_2;Y) = I(X_1,X_2;Y)$.  The proof in the Appendix depends on the InfoMin proposition from Tian et al (2020), that posits good representations also minimize $I(Z_1;Y|Z_2)$.  This is not the same as saying that normal CL approaches necessarily minimize $I(Z_1;Y|Z_2)$.  The identity transform also optimizes Eqn 7, the objectives defining the unique representations, suggesting something else is needed in the objective.
- The idealized augmentation strategy is a nice return to the self-supervised setting, but it feels too far removed from reality. L167: "In the case of image classification, task-relevant information is the object in the picture, while task-irrelevant information is the background."  No, the task relevant information is only a couple of bits, and $I(X;X^\prime)=I(X;Y)$ is not achieved by only swapping out the background.  An augmentation that would achieve this extremely high bar would swap the object for another of its class, changing the image entirely.  Perhaps another way to see it: every pixel pertaining to the car in Fig 4 could be jittered, or the structure of the car could be changed quite significantly while still linking it to the original image X, and this represents a ton of information $I(X;X^\prime)$ for any commonly used augmentation, far exceeding $I(X;Y) \le H(Y)$.  While motivation by an ideal augmentation can be helpful, I think the section would benefit from a more sober take on the implications of realistic augmentations on the proposed information decomposition.
- Important details seem to be missing.
  - What specifically is Cross+Self and Cross+Self+Fact?  It is referred to in vague terms, as a “category” (L219), “capturing a range of methods” (L218), and Cross+Self+Fact “is approximately done in prior work” (L221).
  - I like the setup of the synthetic experiments, but they are opaque as well.  Without more information, the accuracies of Fig. 1 mean little.  All methods achieve better than 90% for all settings, even SimCLR when all information is unique?  The text (main and appendix) is vague: “The label $y$ is generated as a function (with nonlinearity and noise) of varying ratios of $w_s$, $w_1$, and $w_2$ to represent shared and unique task-relevant information” (L239 and 874).  After searching the attached code, the label $Y$ is created with a threshold on a sigmoid of the average of vector components, which is no different than a simple hyperplane -- no nonlinearity and no noise.  With more clarity, the synthetic results could be a much stronger inclusion in the current work.

**Questions:**

- In theory, the interaction information (“task relevant shared information”) $I(X_1;X_2;Y)$ can be negative.  Are there any interesting consequences of such a scenario in the proposed method?
- Is Definition 1 expressing what is intended?  I imagine the intent is to express $I(X_1;Y|X_2)>0$, but it does not seem correct.
- What is RUS?  It is all over the attached code, without any reference in the text nor any comments in the code.  RUSModel, RUSAugment, RUS_CIFAR10, etc.
Minor points:
- There is a $1/K$ missing in the denominator of Eqn. 10
- Typo in Fig 1, both input variables are labeled $X_1$
- Typo in Line 113, the unique information is written as the shared info
- Fig 2: it is confusing that the argmax LHS is replaced with information bounds on the RHS, and that $Y$ is replaced by $X_1^`$ and $X_2^`$ without explanation and without removing $Y$ from the Venn diagrams on the RHS
- Some mutual information quantities seem to be in bits (Table 1) and others in nats (Fig 3), though it’s rarely stated.

**Limitations:**

Yes, limitations were addressed in Appendix A.

---

> ### Author Rebuttal · Authors · 2023-08-10
>
> [Conditional MI] We apologize for the incomplete reference in Appendix and have fixed it. This conditioning scheme is briefly stated in Lines 171-172 and elaborated in Lines 826-830 for supervised and Lines 834-837 for SSL. Conditioning is done by concatenating the encoded $X_1$ and encoded $X_1’$, concatenating encoded $X_2$ and encoded $X_2’$, and feeding the two vectors into $I_{NCE}$ and $I_{NCE-CLUB}$ - see Figure 1 in attached rebuttal pdf. Conditioning by concatenating is commonly used, for example, in:
>
> Mirza and Osindero, Conditional Generative Adversarial Nets. 2014.
>
> Reed et al., Generative Adversarial Text to Image Synthesis. ICML 2016.
>
> Rombach et al., High-Resolution Image Synthesis with Latent Diffusion Models. CVPR 2022.
>
> [Theorem 1] **We would like to clarify three assumptions referred to in this paper, which have subtle differences (using $I(X_1; Y | X_2) $ without loss of generality):**
> 1. Multi-view redundancy (about data): $I(X_1; Y | X_2) \le \epsilon$, used by standard CL but **not** by Theorem 1, states that the task-relevant information from unique part is minimal ($\le \epsilon$);
> 2. Non-negative unique information (about data): $I(X_1; Y | X_2) > 0$, used by Theorem 1 (Line 94), states that task-relevant information from unique part is nonzero;
> 3. InfoMin Tian et al. (about representation): $I(Z_1; Y | X_2) = 0$, used by Theorem 1 (Lines 645-649), states that the optimal representation $Z_1$ learns task-relevant information only from the shared part.
>
> We will clarify these in the main text. **Theorem 1 is correct under Assumptions 2 and 3.** Under Assumptions 2 and 3, $Z_1=X_1, Z_2=X_2$ is not possible for Theorem 1, as $Z_1$ and $Z_2$ only capture the shared part from Assumption 3, but $X_1$ and $X_2$ contain task-relevant information from the unique part from Assumption 2, which cannot be captured by $Z_1$.
>
> **We also check Assumptions 2 and 3 empirically**. Table 1 and 4 in the main text show that Assumption 2 holds empirically. To verify Assumption 3, we use the synthetic dataset as Table 1 and measure $I(Z_1; Y | X_2)$ in standard CL (SimCLR). We get $I(X_1; X_2)=12.29$ and $I(Z_1; Y | X_2)=0.4$ (see Table 1 in the rebuttal pdf). $I(Z_1; Y | X_2)$ is much smaller and closer to zero than $I(X_1; X_2)$, indicating that both Assumptions are reasonable and Theorem 1 holds in practice.
>
> Nevertheless, if the InfoMin assumption does not hold, we get $I(Z_1, Z_2; Y) \leq I(X_1, X_2; Y)$, with the equality satisfied only if $Z_1=X_1$ and $Z_2=X_2$. We will add the equality case to our results. Since identity transformation is nearly impossible in real-world datasets, Theorem 1 still holds for empirical scenarios.
>
> Identity transformation also optimizes the uniqueness in Eq. 7: this does not violate any of our assumptions and results but does not yield ideal representations for each term (Lines 124-129). Empirically, through the term $-I_{NCE-CLUB}(X_1; X_2)$ in Eq. 9, the loss will remove the shared part in $Z_{U_1}$ and $Z_{U_2}$, making $Z_{U_1} = X_1$ and  $Z_{U_2} = X_2$ practically very unlikely.
>
> [Augmentation] Below, we discuss the implications of three types of augmentations:
> 1. Augmentations that exactly satisfy $I(X_1; X_1') = I(X_1; Y)$ and $I(X_2; X_2'|X_1) = I(X_2; Y|X_1)$ (Eqs.17-18); exactly satisfying these conditions is hard, but empirically we do not exactly require this assumption - instead we relax it to:
> 2. Augmentations that approximately satisfy $I(X_1; X_1') \approx I(X_1; Y)$ and $I(X_2; X_2'|X_1) \approx I(X_2; Y|X_1)$: our intuition is to avoid augmentations that will remove or strongly destroy information shared by the other modality (e.g., the caption), and augment via cropping or color jittering in the image. We will clarify this in the paper as a guideline for practitioners. These **consistently outperform independent augmentations (Table 3), suggesting that approximately satisfying this condition is sufficient**.
> 3. Augmentations that do not consider this condition at all (i.e., independent modality augmentations as done in prior work): this has worse performance.
>
> [Cross+self] Cross + Self refers to all self-supervised methods that learn one representation trained jointly for two objectives: cross-modal contrastive loss (e.g., image-text contrastive) plus unimodal contrastive (e.g., vision only contrastive from two image views) [e.g., 27,29,56].
>
> Cross+Self+Fact includes self-supervised methods with separate representations, one trained for cross-modal contrastive objectives and another trained for unimodal contrastive objectives, such as [76,79].
>
> [synthetic exps] The data is generated using multivariate Gaussians with fixed means and variances, which add noise and randomness. The label is a linear function of the latent variables - we found this setup to display the most evident trends. We also tried non-linear data with noise but observed that all methods fluctuate more since we only do linear probing on the final representation. We refer the reader to our experiments on real-world datasets (Table 2) for comparisons with non-linear labels and noise.
>
> [Negative interactive information] Interaction information is:
> 1. Positive when there is more task-relevant shared info than irrelevant, so FactorCL has to do more work in **capturing** task-relevant information (via lower bound).
> 2. Negative when there is more task-irrelevant shared info than relevant, so FactorCL has to do more work in **removing** task-irrelevant information (via upper bound).
>
> These insights also give a better idea of the weights assigned to learning from these two objectives.
>
> [RUS] RUS was the initial acronym we used, standing for redundant, unique, and synergistic interactions that we aimed to learn via factorized learning.
>
> [Typos and bits vs. nats] We have corrected the term and added notes to distinguish when we use nats or bits.
>
> [Figure] We have fixed the typo in Figures 1 and 2 (used $X’$ instead of $Y$) and added them to the rebuttal PDF.

---

> > ### Comment · Reviewer_fNpT · 2023-08-18
> >
> > I am worried that there remains no support for the conditional mutual information estimation that forms such an important part of the proposed method.
> >
> > The three references proposed in the rebuttal are emphatically $\textbf{not}$ estimating conditional mutual information.  Concatenation opens the function space to include distributions conditioned on the concatenated variable, but it does not prescribe a conditional distribution.
> >
> > Consider the schematic in the rebuttal pdf.  InfoNCE lower bounds the mutual information between the random variables serving as its two inputs.  The two inputs are two new random variables formed by the concatenation -- call $\tilde{Z}_1 \sim p(Z_1, Z_1^\prime)$ and similarly for $\tilde{Z}_2$.  InfoNCE provides a lower bound on $I(\tilde{Z}_1;\tilde{Z}_2)$.  Phrased another way, how can InfoNCE know what part of the concatenation to condition on?
> >
> > Conditional mutual information is not trivial to estimate; see for example "CCMI : Classifier based Conditional Mutual Information Estimation", Mukherjee et al. 2020.
> >
> > If the authors can resolve this issue or point out a misunderstanding on my end, I am happy to raise my score.  Otherwise I am leaning more towards reject than before, because the information theoretic basis of this work is called into question.
> >
> > Another thing: the identity transform is just a counterexample.  It represents any invertible transformation, meaning all information is preserved.  This is far more common than the authors suggest in the rebuttal ("Since identity transformation is nearly impossible in real-world datasets, Theorem 1 still holds for empirical scenarios.") -- in general, a point transform between two continuous spaces will preserve all information.  See e.g. "On the information bottleneck theory of deep learning" (Saxe et al. 2018) for a nice discussion around this point.

---

> > > ### Author Response · Authors · 2023-08-20
> > >
> > > We thank the reviewer for the comment, we would like to clarify our estimation of Conditional MI (CMI):
> > >
> > > Consider the CMI term $I(X_1, X_2 | C)$, where $C$ is a conditioning variable. **Conditional InfoNCE (our Eq. 15) is proved to be a lower bound of CMI in Sordoni et al., Proposition 1**:
> > >
> > > $$I_\text{NCE}(X_1; X_2|C) =  \mathbb{E}_{p(C)} \left[ \mathbb{E} \left[ \log \frac{\exp \{\phi(x_1,x_2^+, c)\}}{\frac{1}{k} \sum_k \exp \{\phi(x_1, x_2^-, c)\}} \right] \right],$$
> > >
> > > where the inner expectation is taken w.r.t $\left(x_1,x_2^+ \sim p(x_1,x_2|c), x_2^- \sim p(x_2|c)\right)$.
> > >
> > > It has two key differences from InfoNCE:
> > >
> > > 1. Positive pairs $(x_1,x_2)$ are sampled from $p(x_1,x_2|c)$, and negative pairs $(x_1,x_2^-)$ are sampled from $p(x_2|c)$.
> > >
> > > 2. The critic takes the form $\phi(x1,x2,c)$, which Sordoni et al. implement as $f([x_1,c])^\top g(x_2)$ for trainable encoders $f()$ and $g()$, $[x_1,c]$ denotes concatenation (Sordoni et al. equation 36).
> > >
> > > We implement our method in the same way, with conditioning variable $C=(X_1’, X_2’)$ as augmented modalities:
> > >
> > > 1. Positive pairs $(x_1,x_2)$ are sampled from $p(x_1,x_2|x_1’,x_2’)$, this is effectively the original modality pair $(x_1,x_2)$ and their augmentations $(x_1’,x_2’)$.
> > >
> > > 2. Sampling negative pairs $(x_1,x_2^-)$ from $p(x_2|x_1’, x_2’)$ is nontrivial, especially when $x_1’, x_2’$ are continuous; Sordoni's boosted critic estimation (their section 4.3) addresses this issue, yielding an empirically accurate CMI estimator. Since we focus on scaling our method to real-world datasets, we use this for simplicity (Sordoni also discuss variational approximations and importance sampling as alternative methods).
> > >
> > > 3. Our critic function takes the form $\phi(x_1,x_2,x_1’,x_2’)$, which we implement as $f([x_1,x_1’])^\top g(x_2,x_2’)$ for trainable encoders $f()$ and $g()$, $[x_1,x_1’]$ and $[x_2,x_2]$ denote concatenation, and $f()$ and $g()$ are specific to modalities 1 and 2 respectively, again justified by Sordoni.
> > >
> > > Essentially, the critic knows which variable $C=c$ is being conditioned on because $c$ stays constant while $x_1,x_2$ change. For every $c \in C$, the model implicitly learns a separate InfoNCE, call it InfoNCE($c$) to lower bound $I(X_1; X_2|C=c)$, since $c$ is now a (concatenated) input to InfoNCE. We have added a detailed discussion and reference to Sordoni in the paper.
> > >
> > > Conditional InfoNCE-CLUB (Eq. 16) can be similarly shown to upper bound CMI, by extending Theorem 5 in Appendix C.1 (InfoNCE-CLUB >= MI) with Proposition 1 in Sordoni et al. (plugging in the optimal critics for Conditional InfoNCE).
> > >
> > > **We also ran new experiments verifying that Conditional InfoNCE $<=$ CMI $<=$ Conditional InfoNCE-CLUB**. Following the setup of Mukherjee et al., with our estimators and true CMI estimated on synthetic data with fixing dimension of representation $Z$ and varying samples $n$, and fixing samples $n$ and varying $d_z$. We used our implementations of conditional InfoNCE (Eq.13) and conditional InfoNCE-CLUB (Eq. 14). Results are:
> > >
> > > | Number of samples ($* 10^3$), fix $d_z$=20      | 5 | 10 | 20 | 50 |
> > > | ----------------- | ---- | ---- | ---- | ---- |
> > > | CCMI (MI-Diff + Classifier) | 2.03 | 2.06 | 2.15 | 2.20 |
> > > | Conditional InfoNCE (our lower bound)   | 2.19 | 2.20 | 2.20 | 2.20 |
> > > | Conditional InfoNCE-CLUB (our upper bound) | 3.45 | 3.53 | 2.98 | 2.86 |
> > > | True CMI | 2.32 | 2.32 | 2.32 | 2.32 |
> > >
> > >
> > > | Dimension $d_z$, fix $n=20*10^3$      | 1 | 10 | 20 | 50 | 100 |
> > > | ------------------ | ---- | ---- | ---- | ---- | ---- |
> > > | CCMI (MI-Diff + Classifier) | 2.30 | 2.18 | 2.15 | 1.98 | 1.67 |
> > > | Conditional InfoNCE (our lower bound)   | 2.18 | 2.20 | 2.20 | 2.26 | 2.30 |
> > > | Conditional InfoNCE-CLUB (our upper bound) | 3.70 | 2.95 | 2.98 | 2.79 | 2.86 |
> > > | True CMI | 2.32 | 2.32 | 2.32 | 2.32 | 2.32 |
> > >
> > >
> > > **Our estimators satisfy Conditional InfoNCE $<=$ CMI $<=$ Conditional InfoNCE-CLUB, and are comparable to Mukherjee et al.’s estimator, suggesting that our method yields valid and competitive lower and upper bounds for CMI.**
> > >
> > > Finally, we clarified in the paper that we learn representations by optimizing lower and upper bounds of CMI, and acknowledge that exact CMI estimation is difficult (referencing Sordoni et al., Mukherjee et al., Molavipour et al.)
> > >
> > > [Identity transformation] We agree that an invertible transformation may exist, nevertheless, in contrastive learning, **representation $Z$ is often lower dimensional than high-dimensional raw data $X$ (images/videos)**, making identity and invertible transformations impossible. This implies that Theorem 1 still holds for empirical scenarios, we have also added a discussion of Saxe et al.
> > >
> > > Thank you again for your extremely insightful feedback.
> > >
> > > Sordoni et al. https://arxiv.org/abs/2106.13401
> > >
> > > Mukherjee et al. https://arxiv.org/abs/1906.01824
> > >
> > > Molavipour et al. https://arxiv.org/abs/2006.07225

---

> > > > ### Comment · Reviewer_fNpT · 2023-08-20
> > > >
> > > > Please help me understand:
> > > > 1. Concatenation works if the contrastive set --- each **full** batch of positive and negative samples --- is conditioned on a single outcome of the conditioned variable, here $x_1^\prime, x_2^\prime$.  This would mean an entire batch based around a single paired $x_1, x_2$, with augmentations sampled repeatedly, right?  Please point me to the location in the code where you do this -- it looks like you sample a full batch of images and augment each one, to have one augmentation per image, but I don't want to waste any remaining time in this discussion period parsing.
> > > > 2. Sordoni writes that their boosted critic is not useful as a conditional mutual information estimator -- the critic is shown to be successful, but the actual CMI must be estimated with the importance sampling objective.  Can you explain why this seems to be irrelevant for your purposes?

---

> > > > > ### Author Response · Authors · 2023-08-20
> > > > > **Further clarifications**
> > > > >
> > > > > [CMI] The ideal case of conditional sampling would consider a different conditional distribution $p(x_1, x_2 | x_1’, x_2’)$ for each data pair $x_1’, x_2’$, which is very expensive. As the reviewer suggested, this may be approximated by creating a batch by augmenting a single pair $x_1, x_2$ multiple times. We originally assumed that 1 augmentation per original $x_1,x_2$ was sufficient because we were testing on large multimodal datasets, so did not seek to augment further. Our code easily supports increasing the number of augmentations - we just ran experiments with 64 augmentations per original image-text pair and computing Conditional InfoNCE per pair.
> > > > >
> > > > > With such a sampling scheme, the new result on the Mustard sarcasm detection dataset [10] is $62.3$% compared to our previous $63.4$%, but we did not tune any hyperparameters, and $62.3$% is still better than the baselines SimCLR $47.8$%, Cross+Self $58.7$%, and Cross+Self+Fact $58.7$%. It is likely that the suggested sampling method would further improve performance.
> > > > >
> > > > > [Critic] Yes, we exactly want a successful critic because our goal is to train the critic for representation learning, which we find yields strong downstream results when applied to real-world multimodal tasks. Furthermore, we find that our estimator performs competitively to CMI estimators like CCMI using their synthetic dataset (results in the previous response).
> > > > >
> > > > > We emphasize that our paper proposes a conceptual framework for factorizing multimodal information, and ways to capture unique information for scalable representation learning. We do not claim to do exact CMI estimation, nor that we are better than other CMI estimators. Different CMI estimators can be plugged into our framework - we choose the contrastive ones for their compatibility with multimodal and make approximations to ensure that our method is as scalable as standard CL. We acknowledge that adding importance sampling and conditional sampling will better estimate CMI, and perhaps result in better downstream representations, but will come with a tradeoff in complexity.
> > > > >
> > > > > We have really appreciated these discussions and references and are working on adding these new results and comparisons to our paper.

---

> > > > > > ### Comment · Reviewer_fNpT · 2023-08-21
> > > > > >
> > > > > > Given the back and forth during this discussion period, I think the validity of the CMI estimation in the original submission is tenuous but that the authors are on the right track, and that the empirical successes+other positives of the work outweigh the need for a perfectly justified CMI estimator.  On the premise that the material of this discussion makes it into the final work, I raise my score to 6.

---

### Author Rebuttal · Authors · 2023-08-10

Dear reviewers, we are extremely grateful for your valuable feedback and insightful comments. We are glad that you agree that our results are innovative (859W), original (fNpT), significant (fNpT), and applicable to a broad range of settings in contrastive learning (859W, bfCW, N8Dx, 4hLy). Your concrete suggestions are a valuable step in this direction, and we have revised our submission accordingly to take these into account. In this short note, we summarize the main changes we made to our submission:
1. Figure 1 to clarify the conditioning process of conditional Info-NCE question raised by Reviewer fNpT.
2. Table 1 to verify the InfoMin assumption (Tian et al.) used in Theorem 1 empirically, showing that Theorem 1 holds in practice - standard CL methods struggle to learn unique information.
3. Table 2 shows FactorCL (ours) further improves performance by using the same modalities and architectures as the supervised baselines from Multibench [37], and FactorCL outperforms the reproduced supervised methods.
4. In Table 3, we also added experiments on NYCaps, a new dataset that requires unique information in cartoon images and humorous captions, and we show FactorCL outperforms SimCLR and SupCon.
5. Even though the optimal augmentations are hard to satisfy, in practice, we use these intuitions to guide augmentation design, yielding state-of-the-art results and scales to real-world datasets. Our intuition is to avoid augmentations that remove or strongly destroy information shared by the other modality (e.g., the caption) and instead augment via cropping or color jittering in the image. We will clarify this in the paper as a guideline for practitioners.
6. Finally, we would like to emphasize that all code and data are in the supplementary material, and we plan to release all code and data on GitHub after the review period so details are transparent and reproducible. We have added all requested details regarding the method and experiments to the appendix.

---

### Decision · Program_Chairs · 2023-09-21

**Decision:**

Accept (poster)

**Comment:**

This paper presents a self-supervised multimodal representation learning method to capture both shared and unique information in multiple modalities. The proposed method factorizes task-relevant information into shared and unique parts, learns specific information content through optimizing upper and lower mutual information bounds, and performs multimodal data augmentations to approximate task relevance.

Reviewers appreciated the motivation, the idea, and the novelty of the proposed method, but also raised concerns on the support of the estimation of conditional MI, Theorem 1, the reality of the augmentation and assumptions, the practical implementation, the experimental results, and technical details. The rebuttal addressed most concerns. The AC agreed with the strengths of the proposed method. Reviewers did raise some valuable concerns that should be addressed. The authors are encouraged to make the necessary changes in the camera-ready version.